# Membrane-mediated dimerization potentiates PIP5K lipid kinase activity

Scott D Hansen[1,2]*, Albert A Lee[3,4,5], Benjamin R Duewell[1,2], Jay T Groves[3,4]*

[1]Department of Chemistry and Biochemistry, University of Oregon, Eugene, United States; [2]Institute of Molecular Biology, University of Oregon, Eugene, United States; [3]Department of Chemistry, University of California, Berkeley, Berkeley, United States; [4]California Institute for Quantitative Biosciences, Berkeley, United States; [5]Department of Molecular and Cell Biology, Berkeley, United States

**Abstract** The phosphatidylinositol 4-phosphate 5-kinase (PIP5K) family of lipid-modifying enzymes generate the majority of phosphatidylinositol 4,5-bisphosphate [PI(4,5)$P_2$] lipids found at the plasma membrane in eukaryotic cells. PI(4,5)$P_2$ lipids serve a critical role in regulating receptor activation, ion channel gating, endocytosis, and actin nucleation. Here, we describe how PIP5K activity is regulated by cooperative binding to PI(4,5)$P_2$ lipids and membrane-mediated dimerization of the kinase domain. In contrast to constitutively dimeric phosphatidylinositol 5-phosphate 4-kinase (PIP4K, type II PIPK), solution PIP5K exists in a weak monomer–dimer equilibrium. PIP5K monomers can associate with PI(4,5)$P_2$-containing membranes and dimerize in a protein density-dependent manner. Although dispensable for cooperative PI(4,5)$P_2$ binding, dimerization enhances the catalytic efficiency of PIP5K through a mechanism consistent with allosteric regulation. Additionally, dimerization amplifies stochastic variation in the kinase reaction velocity and strengthens effects such as the recently described stochastic geometry sensing. Overall, the mechanism of PIP5K membrane binding creates a broad dynamic range of lipid kinase activities that are coupled to the density of PI(4,5)$P_2$ and membrane-bound kinase.

*For correspondence:
shansen5@uoregon.edu (SDH);
JTGroves@lbl.gov (JTG)

**Competing interest:** The authors declare that no competing interests exist.

## Editor's evaluation

This study presents highly interesting and detailed biochemical analyses into the mutual relationship between PI(4,5)$P_2$ lipids and their kinase PIP5K, which engage in an exciting pattern-forming reaction on membranes. Using direct single-molecule imaging approaches, the authors find cooperative recruitment of PIP5K to the membrane, dimerization-enhanced catalytic efficiency and indications of allosteric regulation. This article is of interest to a wide range of readers who study the biology of lipid-modifying enzymes, especially as it relates to interfacial reaction kinetics in biological membranes.

## Introduction

Phosphatidylinositol phosphate (PIP) lipids are an important class of second messengers that regulate the localization and activity of proteins on every intracellular membrane in eukaryotic cells (*Di Paolo and De Camilli, 2006*; *Balla, 2013*). Synthesis of PIP lipids is regulated by different classes of lipid kinases and phosphatases that drive the interconversion between PIP lipid species through the phosphorylation and dephosphorylation of inositol head groups. Of particular importance to a vast array of signaling pathways are phosphatidylinositol 4,5-bisphosphate [PI(4,5)$P_2$] lipids, which comprise a minor phospholipid component of the total cellular membrane composition (i.e., ~2 %) (*Wenk et al., 2003*; *Nasuhoglu et al., 2002*). PI(4,5)$P_2$ serves many important functions in biological processes,

including receptor activation, ion channel function (*Hansen, 2015*), endocytosis (*Zoncu et al., 2007*; *Jost et al., 1998*), and actin network assembly at the plasma membrane (*Janmey et al., 2018*). Understanding the mechanisms that control PI(4,5)P$_2$ lipid synthesis is critical for deciphering how cells regulate receptor signaling and PIP lipid homeostasis at the plasma membrane.

Two classes of lipid kinases catalyze the production of PI(4,5)P$_2$ lipids: phosphatidylinositol 4-phosphate 5-kinase (PIP5K, type I PIPK) and phosphatidylinositol 5-phosphate 4-kinase (PIP4K, type II PIPK) (*Burke, 2018*). These two families of lipid kinases differ in substrate specificity; PIP5K phosphorylates phosphatidylinositol 4-phosphate – PI(4)P, while PIP4K phosphorylates phosphatidylinositol 5-phosphate – PI(5)P (*Loijens and Anderson, 1996*; *Rameh et al., 1997*; *Muftuoglu et al., 2016*). Due to the higher abundance of PI(4)P in the plasma membrane, relative to PI(5)P, the majority of PI(4,5)P$_2$ lipids are generated from reactions catalyzed by PIP5K (*Doughman et al., 2003*; *Balla, 2013*). Functionally conserved across eukaryotes, yeast express a single PIP5K enzyme denoted Mss4 (multicopy suppressor of Stt4) (*Yoshida et al., 1994*; *Homma et al., 1998*; *Desrivières et al., 1998*). In humans, three paralogs – PIP5KA, PIP5KB, and PIP5KC – regulate the production of most PI(4,5)P$_2$ lipids found at the plasma membrane.

Several mechanisms regulate membrane docking of PIP5K, including substrate recognition (*Muftuoglu et al., 2016*; *Kunz et al., 2000*), electrostatic interactions with anionic lipids at the cell plasma membrane (*Fairn et al., 2009*), and membrane binding of an amphipathic helix (*Liu et al., 2016*). Single-molecule characterization of human PIP5KB has revealed a role for PI(4,5)P$_2$ lipids in controlling cooperative membrane association and positive feedback during PI(4)P lipid phosphorylation reaction (*Hansen et al., 2019*). Structural and biochemical studies indicate that, like PIP4K (*Rao et al., 1998*; *Burden et al., 1999*), PIP5K can homodimerize in solution (*Hu et al., 2015*). In the case of zebrafish PIP5KA (zPIP5KA), dimerization has been shown to be required for lipid kinase activity (*Hu et al., 2015*). It remains unclear whether dimerization regulates membrane docking, ATP binding, or catalysis of PIP5K. Overall, the sequence of molecular interactions that control PIP5K membrane localization and activation has not been elucidated.

Using total internal reflection fluorescence (TIRF) microscopy to measure the kinetics of PIP lipid phosphorylation on supported lipid bilayers (SLBs) in vitro, we previously reported that human PIP5KB catalyzes the phosphorylation of PI(4)P with a positive feedback loop based on association with its reaction product, PI(4,5)P$_2$ (*Hansen et al., 2019*). Based on the crystal structure of zebrafish PIP5KA (*Hu et al., 2015*; *Muftuoglu et al., 2016*) and previous biochemical data, we have worked under the assumption that members of the PIP5K protein family function as obligate dimers. However, through comparative single-molecule TIRF (smTIRF) microscope measurements of PIP4K and PIP5K membrane-binding dynamics in vitro we discovered that members of the PIP5K protein family exist in a monomer–dimer equilibrium in solution. At low molecular densities, PIP5K protein family members can associate with PI(4,5)P$_2$ membranes as a monomer and catalyze the phosphorylation of PI(4)P. Under these conditions, the mechanism of PIP5K-positive feedback is controlled by cooperative binding to the reaction product, PI(4,5)P$_2$. Increasing the surface density of membrane-bound PIP5K promotes dimerization, which further increases the dwell time and enhances the catalytic efficiency of the kinase ~20-fold. Consistent with a mechanism of allosteric regulation, dimerization can increase PIP5K catalytic efficiency independently of enhancing membrane avidity. We find that the increase in kinase activity afforded by membrane-mediated dimerization – more specifically the strong positive feedback it creates – dramatically enhances the PIP5K's ability to form bistable PIP compositional patterns in the presence of an opposing PIP lipid phosphatase on SLBs. The membrane-mediated dimerization also amplifies stochastic fluctuations in kinase reaction velocity. In the context of spatial confinement, these magnified fluctuations facilitate mechanisms such as the recently reported stochastic geometry sensing, in which bistability and even the deterministic outcome of a competitive reaction may depend on system size (*Hansen et al., 2019*; *Lee et al., 2021*). Together, our results highlight a mechanism by which PI(4,5)P$_2$ binding and membrane-mediated dimerization create a broad dynamic range of PIP5K activities that cells can potentially leverage to tune the concentration and spatial distribution of PI(4,5)P$_2$ lipids on cellular membranes.

## Results

### PIP4K and PIP5K bind to PI(4,5)P$_2$ membranes with distinct oligomerization states

The PIP4K and PIP5K families of lipid kinases both reportedly form homodimeric complexes, but with structurally distinct dimer interfaces (*Figure 1A*; *Hu et al., 2015*; *Rao et al., 1998*; *Burden et al., 1999*). Consistent with size-exclusion chromatography and multi-angle light scattering (SEC/ MALS) data published by *Hu et al., 2015*, we found that wild-type and the dimerization-deficient zebrafish PIP5KA (zPIP5KA) mutant, D84R, showed distinct SEC elution profiles that reflect the dimeric and monomeric states of each kinase (*Figure 1—figure supplement 1*). However, comparing the SEC elution profiles of zPIP5KA and PIP4KB – both predicted to have dimeric molecular weights of 90 kDa – indicated that zPIP5KA eluted slowly compared to PIP4KB and protein molecular weight standards (*Figure 1—figure supplement 1*). Differences in the elution profile of zPIP5KA and PIP4KB could be the result of distinct subunit orientations or the oligomerization state. This observation led us to further investigate whether the PIP5K family of proteins generally exists in a monomer–dimer equilibrium, rather than being obligate dimers like PIP4K. Unfortunately, our attempts to measure the strength of PIP5K dimerization in solution using fluorescence polarization anisotropy and analytical ultracentrifugation were unsuccessful. For this reason, we established a single-molecule imaging approach to directly visualize and quantify the propensity of PIP4K and PIP5K proteins in solution to associate with PI(4,5)P$_2$-containing membranes as either a monomer or dimer. Deciphering the relationship between PIP5K membrane binding, oligomerization, and catalysis, we describe here the molecular basis of positive feedback during PIP5K-dependent generation of PI(4,5)P$_2$ lipids.

To determine whether PIP4K and PIP5K proteins in solution can be recruited to membranes with distinct oligomerization states, we established a single-molecule cell lysate assay (*Lee et al., 2017*) to compare the membrane-binding properties of fluorescently labeled human PIP4KB and PIP5KB (referred to as 4KB and 5KB in figures) on SLBs using smTIRF microscopy. For these experiments, genes encoding mNeonGreen (mNG) fused to either PIP4KB or PIP5KB were transiently expressed in human embryonic kidney (HEK) 293 cells. Cells were lysed by sonication and centrifuged to remove membranes and debris. We quantified the concentration of mNG-PIP4KB and mNG-PIP5KB in clarified cell lysate using a purified mNG protein standard (*Figure 1—figure supplement 2*). Samples containing mNG-labeled kinase were then diluted ~10,000-fold in imaging buffer to a concentration of 10 pM and incubated on supported membranes containing 4% PI(4,5)P$_2$ lipids. Under these conditions, the resulting surface density of membrane-bound mNG-PIP4KB and mNG-PIP5KB was ~0.03 molecules/μm$^2$ (or ~100 molecules per field of view) (*Figure 1B*).

To quantify the differences in the oligomerization states of membrane-bound mNG-PIP4KB and mNG-PIP5KB, we compared the molecular brightness and diffusion coefficients of single particles by smTIRF microscopy. Intensity line scans through mNG foci revealed that the majority of mNG-PIP4KB molecules were two times brighter compared to membrane-bound mNG-PIP5KB (*Figure 1C*). Because mNG-PIP4KB is a obligate dimer (*Rao et al., 1998*; *Burden et al., 1999*), the molecular brightness distribution of this lipid kinase set the upper limit for the percentage of detectable dimers in our assay. This was based on the fraction of mNG molecules that formed mature chromophores during expression in HEK293 cells, which was ~80% for mNG-PIP4KB. To determine the oligomerization state of membrane-bound mNG-PIP4KB and mNG-PIP5KB, we performed single-molecule photobleaching experiments. In the case of mNG-PIP4KB, the vast majority of membrane-bound particles photobleached in two steps (*Figure 1D*). By contrast, single-membrane-bound mNG-PIP5KB particles photobleached in a single step (*Figure 1E*).

To measure changes in the molecular brightness distribution, we continuously monitored the fluorescence of either membrane-bound mNG-PIP4KB or mNG-PIP5KB under conditions that promoted stepwise photobleaching. Over the course of a 10 s photobleaching experiment, the molecular brightness distribution for mNG-PIP4KB shifted from bimodal to a single peak (*Figure 1F*, *Figure 1—video 1*). During this time, the fraction of mNG-PIP4KB dimers with two visible mNG fluorophores gradually decreased from ~80% to ~20% of the population (*Figure 1H*). By contrast, the molecular brightness distribution of mNG-PIP5KB remained unimodal throughout the 10 s photobleaching experiment, consistent with mNG-PIP5KB associating with PI(4,5)P$_2$-containing membranes predominantly as a monomer at a low protein surface density (*Figure 1G*, *Figure 1—video 2*). We also observed similar differences in molecular brightness comparing recombinantly expressed and purified PIP4KB

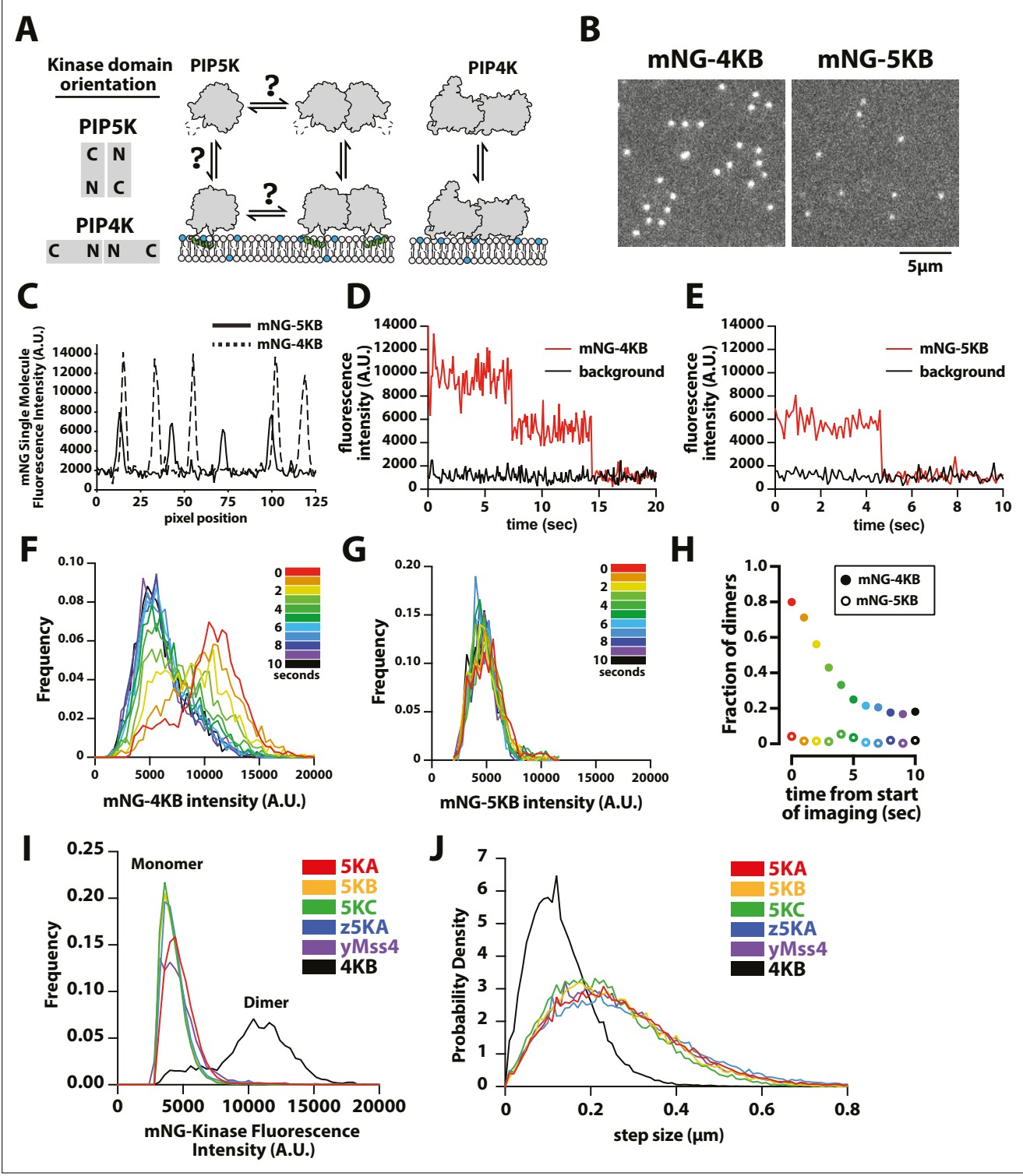

**Figure 1.** PIP4K and PIP5K can bind to PI(4,5)P$_2$ membranes with distinct oligomeric states. (**A**) Cartoon showing the kinase domain orientation and proposed oligomerization states of PIP4K and PIP5K homodimers. PIP5K potentially exists in a monomer–dimer equilibrium, while PIP4K is a constitutive dimer. (**B**) Single-molecule total internal reflection fluorescence (TIRF) microscopy images of supported lipid bilayers (SLBs) incubated with either 1 pM mNG-PIP4KB and 5 pM mNG-PIP5KB. (**C**) Intensity line scans through single mNG-PIP4KB and mNG-PIP5KB molecules bound to membranes in (**B**). (**D, E**) Single-molecule photobleaching dynamics of mNG-PIP4KB (red line) and mNG-PIP5KB (red line) compared to background fluorescence (black line). (**F, G**) Dynamic of change in molecular brightness frequency distribution of mNG-PIP4KB and mNG-PIP5KB during photobleaching. Each histogram represents the population distribution measured during a 1 s window over the course of a 10 s photobleaching experiment. N = 1300–1500 mNG-

*Figure 1 continued on next page*

*Figure 1 continued*

PIP4KB particles per second (**F** and *Figure 1—video 1*), N = 200–300 mNG-PIP5KB particles per second (**G** and *Figure 1—video 1*). (**H**) Probability of observing mNG-PIP4KB and mNG-PIP5KB dimers at different times during the photobleaching experiment shown in (**F, G**). The threshold particle intensity was ≥8200 (A.U.) to be considered a dimer from data in (**F, G**). Note that the fraction of mNG-PIP4KB dimers does not approach 0% because there are new membrane-binding events during the image acquisition. (**I**) Molecular brightness distributions measured in the presence of mNG-tagged mouse PIP5KA (5KA), human PIP5KB (5KB), human PIP5KC (5KC), zebrafish PIP5KA (z5KA), yeast Mss4 (yMss4), and human PIP4KB (4KB) (N = 3911–23304 particles per distribution). (**J**) Step-size distribution plots generated from single-molecule tracking in the presence of mNG-tagged 5KA (D = 0.214 $\mu m^2$/s), 5KB (D = 0.191 $\mu m^2$/s), 5KC (D = 0.165 $\mu m^2$/s), z5KA (D = 0.234 $\mu m^2$/s), yMss4 (D = 0.199 $\mu m^2$/s), and 4KB (D = 0.051 $\mu m^2$/s) (N > 10,000 total steps [or displacements in μm] per distribution derived from 2 to 3 technical replicates). (**B–J**) Membrane composition: 96% DOPC, 4% PI(4,5)P$_2$. Note that the percentage of mature and fluorescent mNG fusions is ~80% for each labeled kinase.

The online version of this article includes the following video, source data, and figure supplement(s) for figure 1:

**Source data 1.** Related to *Figure 1C.*

**Source data 2.** Rlated to *Figure 1D.*

**Source data 3.** Related to *Figure 1E.*

**Source data 4.** Related to *Figure 1F.*

**Source data 5.** Related to *Figure 1G*.

**Source data 6.** Related to *Figure 1H*.

**Source data 7.** Related to *Figure 1I*.

**Source data 8.** Related to *Figure 1J*.

**Figure supplement 1.** Size-exclusion chromatography (SEC) analysis of purified proteins.

**Figure supplement 1—source data 1.** Related to *Figure 1—figure supplement 1*.

**Figure supplement 2.** mNeonGreen calibration curve for measuring the solution concentration in cell lysate.

**Figure supplement 2—source data 1.** related to *Figure 1—figure supplement 2*.

**Figure supplement 3.** Molecular brightness distribution of Ax488-PIP4KB and Ax488-PIP5KB.

**Figure supplement 3—source data 1.** Related to *Figure 1—figure supplement 3*.

**Figure supplement 4.** Molecular brightness and photobleaching analysis of mNG-PIP4K and mNG-PIP5K.

**Figure supplement 4—source data 1.** Related to *Figure 1—figure supplement 4*.

**Figure 1—video 1.** Membrane -binding dynamics and multistep photobleaching of mNG-PIP4KB visualized by single-molecule total internal reflection fluorescence (smTIRF) microscopy.
https://elifesciences.org/articles/73747/figures#fig1video1

**Figure 1—video 2.** Membrane -binding dynamics and single -step photobleaching of mNG-PIP5KB visualized by single-molecule total internal reflection fluorescence (smTIRF) microscopy.
https://elifesciences.org/articles/73747/figures#fig1video2

and PIP5KB that were chemically labeled with Alexa488 in vitro using Sortase-mediated peptide ligation (*Figure 1—figure supplement 3*). However, the lower labeling efficiency achieved using Sortase-mediated peptide ligation led us to measure the molecular brightness distribution of PIP5K homologs and paralogs using exclusively mNG fusion proteins in this study. Comparing the molecular brightness distributions of mNG-PIP5KA (mouse), mNG-PIP5KB (human), mNG-PIP5KC (human), mNG-zPIP5KA (zebrafish), and mNG-yMss4 (yeast multicopy suppressor of Stt4 or yeast phosphatidylinositol 4-phosphate 5-kinase), we also observed only single peaks corresponding to monomeric kinases labeled with a single fluorescent mNG (*Figure 1I*). Photobleaching analysis of membrane-bound mNG-PIP5KA and mNG-zPIP5KA produced similar single-step photobleaching results to those observed for mNG-PIP5KB (*Figure 1—figure supplement 4*).

Consistent with mNG-PIP5K and mNG-PIP4K binding to membranes as either monomers and dimers, respectively, the two lipid kinases also exhibited strikingly different diffusivity when membrane bound. At low surface densities (~0.01 molecules/$\mu m^2$), the mobility of mNG-PIP4KB (0.04 $\mu m^2$/s) was much slower compared to mNG-PIP5K homologs and paralogs (0.15–0.18 $\mu m^2$/s) (*Figure 1J*). These diffusion coefficients were very similar to that measured for PIP5KB labeled with an Alexa647 chemical dye (*Figure 2C*), indicating that membrane binding of the mNG kinases in dilute cell lysate is indistinguishable from recombinantly purified and fluorescently labeled enzymes.

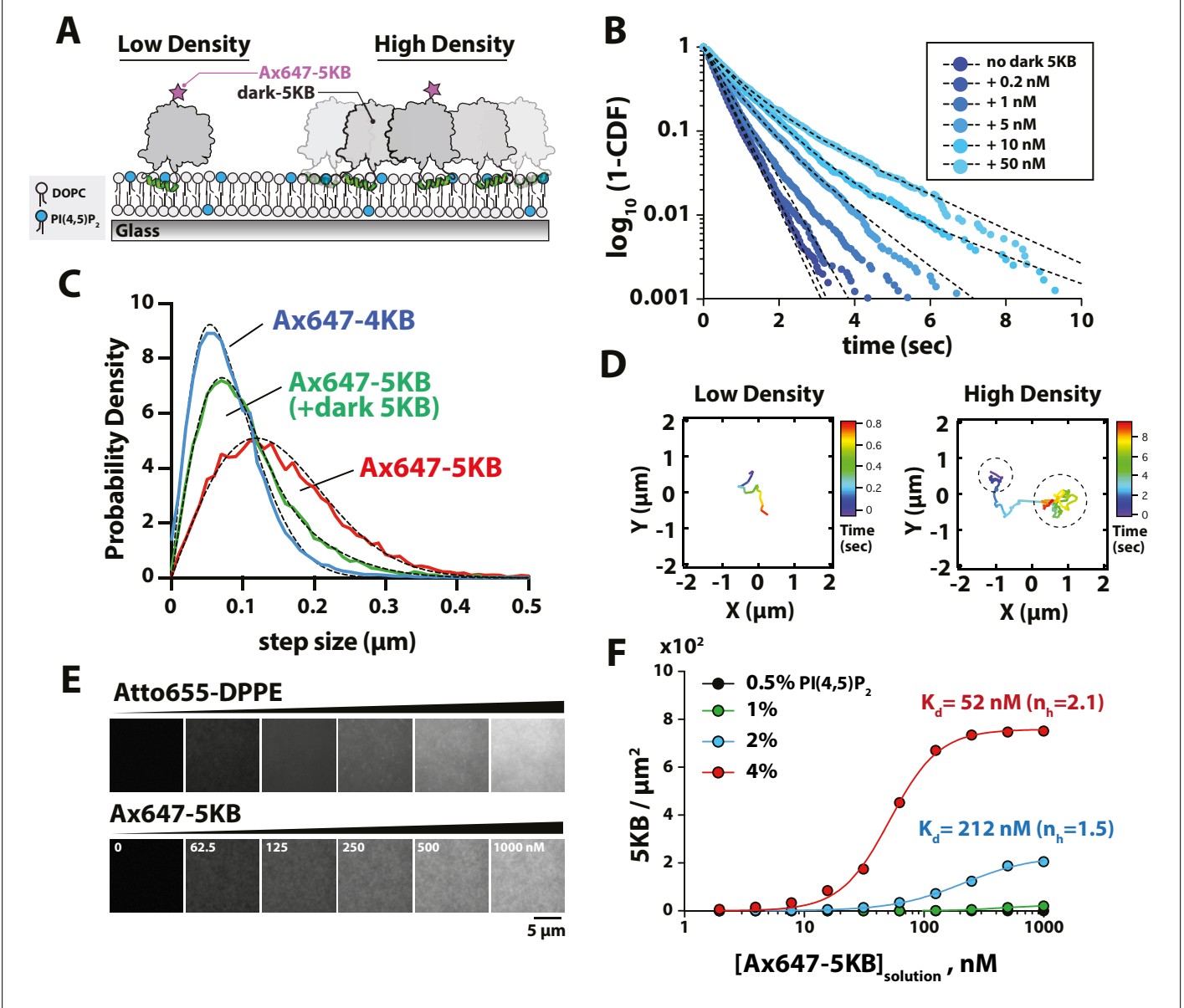

**Figure 2.** Protein density-dependent changes in PIP5K membrane binding. (**A**) Supported lipid bilayer assay for measuring the single-molecule membrane-binding behavior Ax647-PIP5KB at low and high membrane surface densities of PIP5KB. Note that the 'high-density' kinase organization does not imply a specific oligomerization state. (**B**) Single-molecule dwell times of Ax647-PIP5KB measured in the presence of increasing concentrations of unlabeled PIP5KB (0–50 nM). Ax647-PIP5K dwell time was calculated by fitting $\log_{10}$(1 – cumulative distribution frequency [CDF]) to either a single- or double-exponential decay curve (black dashed lines). Bin size equals 50 ms. See *Table 1* for statistics. (**C**) Representative step-size distributions measured in the presence of either 1 pM Ax647-PIP5KB (red), 1 pM Ax647-PIP5KB + 50 nM PIP5KB (green), or 1 pM Ax647-PIP4KB (blue). Dashed black line represents the curve fit used to calculate the diffusion coefficient (see 'Materials and methods'). See *Table 1* for statistics. (**D**) Representative trajectories showing the time-dependent movement of a single-membrane-bound Ax647-PIP5KB (1 pM) in the absence or presence of 5 nM dark PIP5KB. (**B–D**) Membrane composition: 98% DOPC, 2% PI(4,5)P$_2$. (**E**) Montage of images showing supported membranes with increasing densities of Atto655 lipids used to calibrate the molecular density of membrane-bound Ax647-PIP5KB. Membrane composition: 96% DOPC, 4% PI(4,5)P$_2$. (**F**) Ax647-PIP5KB binds cooperatively to membranes containing PI(4,5)P$_2$ lipids. The density of membrane-bound PIP5KB was measured in the presence of increasing solution concentrations of Ax647-PIP5KB on membranes containing 0.5, 1, 2, or 4% PI(4,5)P$_2$ lipids. Lines represent curve fit using concerted model for cooperativity (i.e., Hill equation). $n_H$ is the Hill coefficient. Points are mean values (N = 15–20 fluorescent intensity measurements per sample from one technical replicate).

The online version of this article includes the following video, source data, and figure supplement(s) for figure 2:

**Source data 1.** Related to *Figure 2B*.

*Figure 2 continued on next page*

*Figure 2 continued*

**Source data 2.** Related to *Figure 2C*.

**Source data 3.** Related to *Figure 2F*.

**Figure supplement 1.** Quantification of Alexa647-PIP5KB photobleaching kinetics.

**Figure supplement 1—source data 1.** Related to *Figure 2—figure supplement 1*.

**Figure supplement 2.** Protein density dependent changes in Ax647-PIP5K single molecule dwell time distributions.

**Figure supplement 2—source data 1.** *Figure 2—figure supplement 2*.

**Figure supplement 3.** Calibration of Alexa647-PIP5KB membrane surface density measurements.

**Figure supplement 3—source data 1.** Related to *Figure 2—figure supplement 3*.

**Figure 2—video 1.** Membrane -binding dynamics of 1 pM Ax647-PIP5KB visualized by single-molecule total internal reflection fluorescence (smTIRF) microscopy.

https://elifesciences.org/articles/73747/figures#fig2video1

**Figure 2—video 2.** Membrane -binding dynamics of 1 pM Ax647-PIP5KB plus 50 nM dark PIP5KB visualized by single-molecule total internal reflection fluorescence (smTIRF) microscopy.

https://elifesciences.org/articles/73747/figures#fig2video2

## Membrane-mediated dimerization of PIP5K

The ability of PIP5K proteins to associate with $PI(4,5)P_2$-containing membranes as monomers at low molecular densities (~0.01 PIP5K/$\mu m^2$) confirms that PIP5K is predominantly monomeric when diluted to low solution concentrations. This supports a model that PIP5K exists in a weak monomer–dimer equilibrium in solution. However, membrane association reduces the effects of translational and rotational entropy, both of which oppose dimerization in solution, enabling even weakly dimerizing species in solution to robustly dimerize on membranes (*Chung et al., 2018*; *Lin et al., 2014*). Thus, we anticipated that membrane binding of PIP5K would favor dimerization at some threshold membrane surface density. In order to characterize PIP5K dimerization as a function of membrane surface density, we performed single-molecule-tracking experiments in the presence of low (~0.01 molecules/$\mu m^2$) and high (~100 molecules/$\mu m^2$) densities of membrane-bound PIP5KB (*Figure 2A*). These measurements reveal molecular binding dwell time on the membrane as well as diffusive mobility, both of which are affected by dimerization. For the single-molecule dwell time measurements, we switched to using the more photostable Alexa647 dye (referred to as Ax647) conjugated to either PIP4KB or PIP5KB ($\tau_{bleach}$ = 26.7 s, *Figure 2—figure supplement 1*). In the presence of a low density of noninteracting Ax647-PIP5KB monomers (i.e., 1–5 pM solution concentration), the distribution of single-molecule dwell times could be fit to a single exponential with a characteristic dwell time of ~0.45 s (*Figure 2B*, *Figure 2—figure supplement 2*, *Figure 2—video 1*). When we increased the solution concentration of dark unlabeled PIP5KB, we observed a second population of long dwelling Ax647-PIP5KB molecules (*Figure 2B*, *Table 1*, *Figure 2—video 2*). At concentrations greater than 1 nM dark unlabeled

**Table 1.** Protein density dependent changes in Ax647-PIP5KB membrane binding behavior.

| Protein visualized | [5KB] | $\tau_1$ ± SD (s) | $\tau_2$ ± SD (s) | $\alpha$ ± SD | N | $D_1$ ± SD ($\mu m^2$/s) | $D_2$ ± SD ($\mu m^2$/s) | $\alpha$ ± SD | Steps |
|---|---|---|---|---|---|---|---|---|---|
| 1 pM Ax647-5KB | 0 | 0.453 ± 0.013 | – | – | 7018 | 0.148 ± 0.005 | – | – | 66,889 |
| 1 pM Ax647-5KB | 0.2 nM | 0.475 ± 0.037 | – | – | 4855 | 0.058 ± 0.011 | 0.194 ± 0.012 | 0.20 ± 0.06 | 48,888 |
| 1 pM Ax647-5KB | 1 nM | 0.559 ± 0.009 | – | – | 4783 | 0.038 ± 0.001 | 0.189 ± 0.004 | 0.27 ± 0.02 | 56,027 |
| 1 pM Ax647-5KB | 5 nM | 0.599 ± 0.055 | 1.354 ± 0.32 | 0.72 ± 0.17 | 4268 | 0.038 ± 0.002 | 0.173 ± 0.007 | 0.40 ± 0.04 | 66,680 |
| 1 pM Ax647-5KB | 10 nM | 0.895 ± 0.032 | 2.791 ± 0.27 | 0.94 ± 0.03 | 2362 | 0.039 ± 0.004 | 0.164 ± 0.002 | 0.48 ± 0.05 | 47,103 |
| 1 pM Ax647-5KB | 50 nM | 0.787 ± 0.057 | 2.08 ± 0.078 | 0.70 ± 0.03 | 1530 | 0.043 ± 0.002 | 0.142 ± 0.008 | 0.63 ± 0.03 | 35,678 |
| | | | | | | | | | |
| 1 pM Ax647-4KB | 0 | | | | | 0.021 ± 0.005 | 0.058 ± 0.007 | 0.39 ± 0.15 | 227,275 |

SD, standard deviation from 3 to 5 technical replicates; N, total number of molecules tracked in 3–5 technical replicates; steps, total number of particle displacements measured in 3–5 technical replicates; alpha, fraction of molecules that have the characteristic dwell time or diffusion coefficient ($\tau_1$ or $D_1$), $\tau_{bleaching}$, Ax647-5KB = 26.7 s (see **Figure 2—figure supplement 1**); membrane composition, 98% DOPC, 2% $PI(4,5)P_2$.

PIP5KB, the resulting dwell time distributions for Ax647-PIP5KB were best fit to a two-species model with two characteristic dwell times (*Figure 2B*, *Table 1*). In addition to observing an enhancement in the dwell time, we also observed a protein surface density-dependent decrease in the diffusion coefficient of membrane-bound Ax647-PIP5KB (*Figure 2C*, *Table 1*). Two-dimensional mobility on the membrane has previously been used as a highly effective measure of membrane surface dimerization reactions (*Chung et al., 2018*; *Chung et al., 2019*). A two-species model was required to fit the step-size distribution of membrane-bound Ax647-PIP5KB in the presence of high kinase density, which was similar to the step-size distribution of Ax647-PIP4KB measured at low molecular densities (*Figure 2C*). Examples of Ax647-PIP5KB molecules that transitioned between slow and fast diffusive states could also be seen when we inspected trajectories at an intermediate kinase density that favored PIP5K dimerization (*Figure 2D*).

To quantify how membrane binding of PIP5KB changes as a function of the PI(4,5)$P_2$ lipid density and the kinase solution concentration, we used a method previously established by *Galush et al., 2008* to measure the membrane surface density of Ax647-PIP5KB compared to a fluorescent lipid standard. Using defined molar concentrations of Atto655-1,2-dipalmitoyl-sn-glycero-3-phosphoethanolamine (Atto655-DPPE) lipids incorporated into supported membranes, we calibrated the fluorescence intensity in order to measure the surface density of membrane-bound Ax647-PIP5KB (*Figure 2E*, *Figure 2—figure supplement 3*). This approach overcame the challenges associated with using fluorescence correlation spectroscopy (FCS) to quantify the high membrane surface density of slow-diffusing Ax647-PIP5KB. These measurements revealed two nonlinear membrane-binding behaviors of Ax647-PIP5KB. First, the density of membrane-bound Ax647-PIP5KB dramatically increased as a function of the PI(4,5)$P_2$ density (*Figure 2F*). Second, increasing the solution concentration promoted cooperative membrane binding of Ax647-PIP5KB based on elevated protein densities (*Figure 2F*). Fitting the membrane-binding curves with a concerted model for cooperativity yielded dissociation constants of 212 nM and 52 nM for Ax647-PIP5KB in the presence of 2% and 4% PI(4,5)$P_2$ lipids, respectively (*Figure 2F*). To determine whether the density-dependent change in the dwell time and the step-size distributions of Ax647-PIP5KB were dependent on dimerization of the kinase domain, we sought to characterize a mutant that disrupts the PIP5KB dimer interface.

Inspection of the zPIP5KA crystal structure (*Hu et al., 2015*) and primary amino acid sequence alignment revealed a high degree of conservation between PIP5K homologs and paralogs (*Figure 3A and B*). Based on conservation of the primary amino acid sequence, we mutated the dimer inter-face of human PIP5KB to make the kinase constitutively monomeric. Using smTIRF microscopy, we compared the dwell times and diffusion coefficients of Ax647-PIP5KB and Ax647-PIP5KB (D51R) under conditions with either low and high protein densities on SLBs. When measured at a low protein surface density (~0.01 molecule/μm²), the PIP5KB (D51R) mutant still bound cooperatively to PI(4,5)$P_2$ lipids in a manner that was indistinguishable from the wild-type kinase (*Figure 3C and D*, *Table 2*). Under these conditions, the diffusion coefficients of membrane-bound Ax647-PIP5KB and Ax647-PIP5KB (D51R) were indistinguishable (*Figure 3—figure supplement 1*). In contrast, single-molecule membrane-binding experiments performed using a high kinase surface density (~100 molecules/μm²) revealed that the single-molecule dwell time of Ax647-PIP5KB increased, while the dwell time of the D51R mutant remained unchanged in the presence of 50 nM unlabeled PIP5KB (*Figure 3E*, *Table 2*). In addition, the diffusion coefficient of wild-type Ax647-PIP5KB decreased due to membrane-mediated dimerization, while diffusivity of Ax647-PIP5KB (D51R) remained unchanged in the presence of 50 nM dark PIP5KB (*Figure 3F*).

Having established that the density-dependent changes in Ax647-PIP5KB membrane-binding behavior are mediated by dimerization, we sought to establish conditions to directly visualize dimerization of membrane-bound PIP5KB with single-molecule resolution. For these experiments, we recombinantly expressed and purified mNG-PIP5KB and mNG-PIP5KB (D51R). Based on the SEC elution profiles, mNG-PIP5KB and mNG-PIP5KB (D51R) adopted distinct oligomerization states when loaded at a solution concentration of 50 μM (*Figure 4A*). This concentration is approximately 1000× greater compared to the reported cellular concentration of PIP5K (*Hein et al., 2015*). Next, we performed single-molecule imaging to compare the molecular brightness of mNG-PIP5KB and mNG-PIP5KB (D51R) *immediately following attachment* to membranes containing 4% PI(4,5)$P_2$ lipids (*Figure 4B*). In the presence of 100 pM solution concentration, the molecular brightness distributions of mNG-PIP5KB and mNG-PIP5KB (D51R) were indistinguishable (*Figure 4C and D*). Under these conditions,

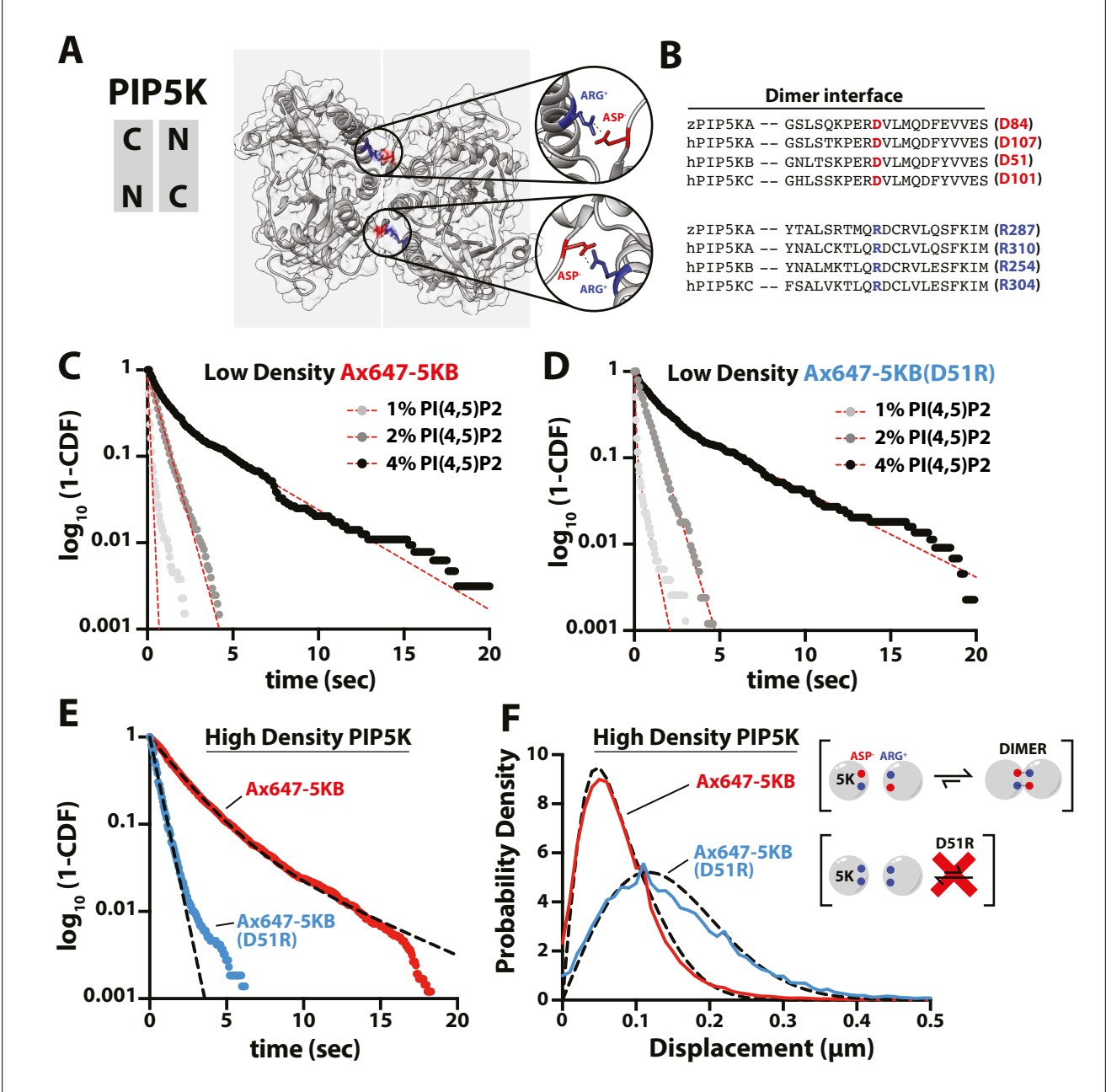

**Figure 3.** PIP5K binds cooperatively to PI(4,5)P$_2$ independent of dimerization. (**A**) Kinase domain orientation for the zebrafish PIP5KA homodimer (4TZ7. pdb). Salt bridges formed between Asp and Arg side chains are colored and shown in zoomed images. (**B**) Sequence alignment between zPIP5KA, human PIP5KA, human PIP5KB, and human PIP5KC highlights conservation of residues in dimer interface. (**C, D**) Dimerization is not required for cooperative PI(4,5)P$_2$ binding. Representative single-molecule dwell time distributions measured at low protein density in the presence of either (**C**) 1–5 pM Ax647-PIP5KB and (**D**) 1–5 pM Ax647-PIP5KB (D51R). Membrane composition: 1, 2, or 4% PI(4,5)P$_2$, plus 96–99% DOPC. (**E, F**) Single-molecule dwell times and step-size distributions measured at 'high density' in the presence of 50 nM non-fluorescent PIP5KB. (**E**) High protein surface density of PIP5KB increases the dwell time of Ax647-PIP5KB, but not Ax647-PIP5KB (D51R). (**F**) Membrane-mediated dimerization is responsible for the protein density-dependent decrease in Ax647-PIP5KB diffusion coefficient. (**C–F**) Membrane composition: 96% DOPC, 4% PI(4,5)P$_2$. Dashed line represents fits of the dwell time distributions. See *Table 2* for statistics.

The online version of this article includes the following source data and figure supplement(s) for figure 3:

**Source data 1.** Related to *Figure 3C*.

**Source data 2.** Related to *Figure 3D*.

**Source data 3.** Related to *Figure 3E*.

*Figure 3 continued on next page*

*Figure 3 continued*

**Source data 4.** Related to *Figure 3F*.

**Figure supplement 1.** Dimer interface mutation does not alter diffusion of Ax647-PIP5K at low protein densities.

**Figure supplement 1—source data 1.** Related to *Figure 3—figure supplement 1*.

both kinases in solution associated with membranes exclusively as monomers. Next, we compared the fluorescence intensity of mNG-PIP5KB and mNG-PIP5KB (D51R) *already membrane bound* and potentially associated with other membrane-bound kinases. For this analysis, we compared the molecular brightness distributions mNG-PIP5KB and mNG-PIP5KB (D51R) at identical membrane surface densities of ~600 mNG-labeled kinases per field of view (i.e., 0.2 kinases/μm²). Under these conditions, we observed two classes of particles with distinct fluorescence intensities (*Figure 4E and F*). Compared to mNG-PIP5KB (D51R), there was a fivefold higher frequency of observing mNG-PIP5KB dimers (*Figure 4G and H*). Photobleaching analysis of the bright mNG-PIP5KB particles revealed a stepwise decrease in fluorescence consistent with a dimer (*Figure 4I*). In our cell lysate experiments, we also observed similar differences in molecular brightness comparing mNG-PIP5KA and mNG-zPIP5KA to their respective dimer interface mutants, mNG-PIP5KA (D92R) and mNG-zPIP5KA (D84R) (*Figure 4— figure supplement 1*).

To directly visualize the association and dissociation dynamics of mNG-PIP5KB dimers, we tracked membrane-bound kinases with high temporal resolution. Under these imaging conditions, we observed a small fraction membrane-bound mNG-PIP5KB monomer diffuse and merge together, forming dimers that persist for 1–2 s before photobleaching or dissociating back into monomers (*Figure 4J*, *Figure 4—video 1*). These dimerization dynamics were not observed for mNG-PIP5KB (D51R). Considering the high membrane surface density used for these experiments, the small fraction of apparent mNG-PIP5KB (D51R) 'dimers' (i.e., 0.9%) represents kinases that transiently cross paths on the membrane without forming a stable dimer complex. Overall, these results support a model in which mNG-PIP5KB can exist as a monomer in solution. However, membrane binding and lateral diffusion promotes membrane-mediated dimerization in a density-dependent manner.

## Dimerization potentiates PIP5K activity independent of enhancing membrane avidity

Dimerization of zPIP5KA was previously shown to be required for lipid kinase activity in vitro (*Hu et al., 2015*). Comparing the kinetics of PI(4,5)P$_2$ production on SLBs in the presence of either PIP5KB and PIP5KB (D51R) revealed that dimerization enhances catalytic activity, but is not essential for generation of PI(4,5)P$_2$ (*Figure 5A and B*). A similar reduction in catalytic activity was observed for mutations that disrupt dimerization of human PIP5KA (*Figure 5—figure supplement 1*). Previous characterization of a dimerization-deficient zPIP5KA, D84R, using circular dichroism demonstrated that disruption of the dimerization interface does not alter protein folding (*Hu et al., 2015*). To confirm that the

**Table 2.** Dimerization dependent changes in Ax647-PIP5KB membrane binding behavior.

| Protein visualized | [5KB] | % PIP$_2$ | $\tau_1$ ± SD (s) | $\tau_2$ ± SD (s) | $\alpha$ ± SD | N | D$_1$ ± SD (μm²/s) | D$_2$ ± SD (μm²/s) | $\alpha$ ± SD | Steps |
|---|---|---|---|---|---|---|---|---|---|---|
| 25 pM Ax647-5KB | 0 | 1 | 0.096 ± 0.05 | – | – | 3422 | – | – | – | – |
| 5 pM Ax647-5KB | 0 | 2 | 0.64 ± 0.092 | – | – | 4525 | 0.171 ± 0.006 | – | – | 174,748 |
| 2 pM Ax647-5KB | 0 | 4 | 0.74 ± 0.088 | 3.43 ± 0.29 | 0.59 ± 0.06 | 1920 | – | – | – | – |
| 25 pM Ax647-5KB (D51R) | 0 | 1 | 0.090 ± 0.01 | – | – | 4848 | – | – | – | – |
| 5 pM Ax647-5KB (D51R) | 0 | 2 | 0.649 ± 0.043 | – | – | 2542 | 0.174 ± 0.001 | – | – | 238,548 |
| 2 pM Ax647-5KB (D51R) | 0 | 4 | 0.864 ± 0.077 | 3.97 ± 0.37 | 0.58 ± 0.03 | 1850 | – | – | – | – |
| 1 pM Ax647-5KB | 50 nM | 2 | 0.709 ± 0.24 | 2.49 ± 0.27 | 0.53 ± 0.08 | 9693 | 0.022 ± 0.002 | 0.067 ± 0.005 | 0.50 ± 0.07 | 311,198 |
| 1 pM Ax647-5KB (D15R) | 50 nM | 2 | 0.416 ± 0.02 | – | – | 8218 | 0.133 ± 0.008 | – | – | 74,928 |

SD, standard deviation from 3 to 5 technical replicates; N, total number of molecules tracked in 3–5 technical replicates; steps, total number of particle displacements measured in 3–5 technical replicates; alpha, fraction of molecules that have the characteristic dwell time or diffusion coefficient ($\tau_1$ or D$_1$), $\tau_{bleaching}$, Ax647-5KB = 26.7 s (see *Figure 2—figure supplement 1*); membrane composition, 96–99% DOPC, 1–4% PI(4,5)P$_2$.

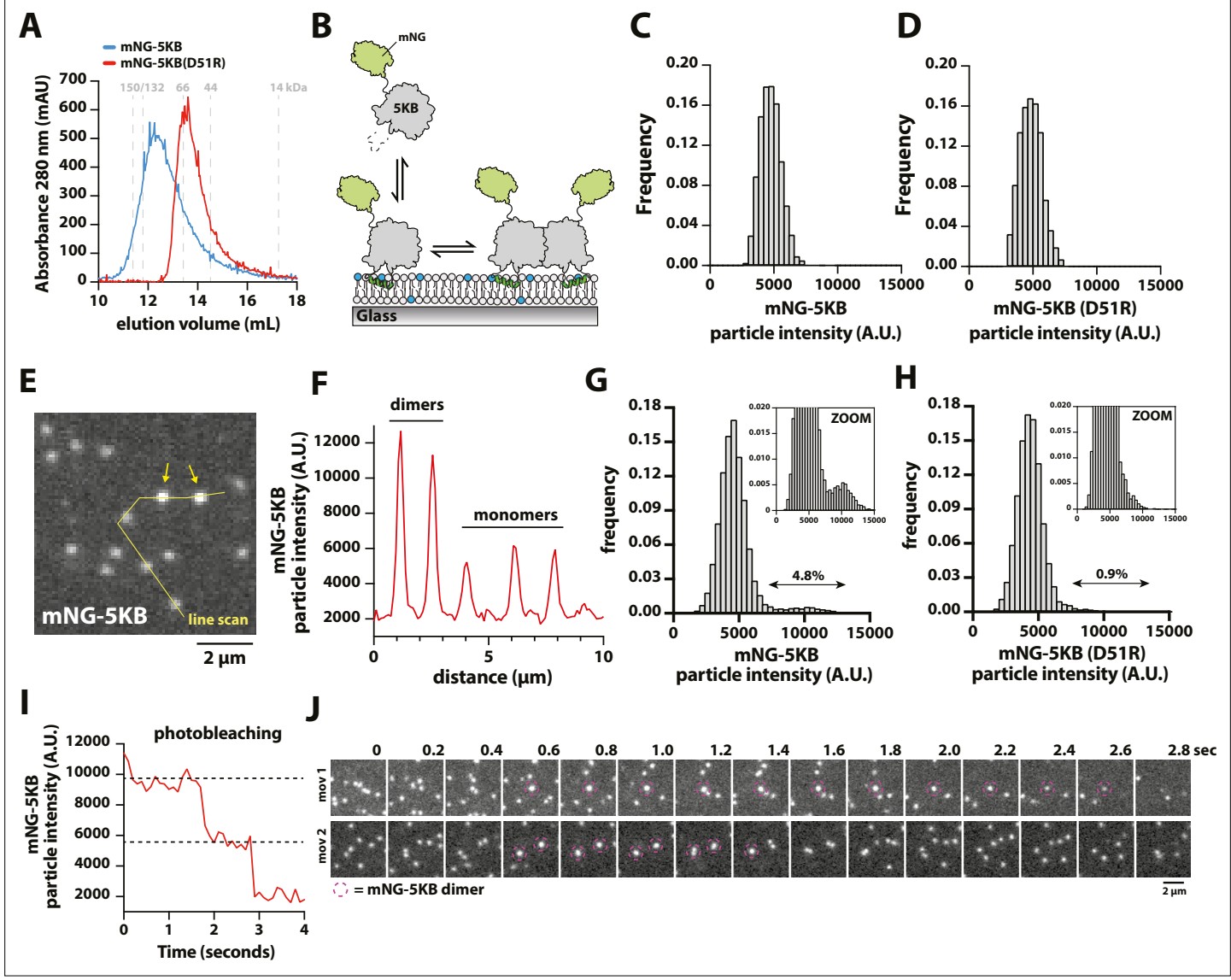

**Figure 4.** Direct visualization of mNG-PIP5KB membrane-mediated dimerization. (**A**) Size-exclusion chromatography (SEC) elution profiles of purified mNG-PIP5KB and mNG-PIP5KB (D51R). The monomeric molecular weight of each kinase equals 75 kDa. Protein was at a concentration of 50 μM (~2 mg total) loaded on a Superdex 200 column. Based on the A280, peak elution concentration is 10–20 μM. (**B**) Diagram showing the experimental setup for visualizing membrane binding of mNG-PIP5KB (WT and D51R). (**C–**, **D**) Molecular brightness distributions of mNG-PIP5KB and mNG-PIP5KB (D51R) based on brightness of single kinases first attaching to the supported membrane (N = 2424 total binding events, four technical replicates in **C**; N = 1747 total binding events, four technical replicates in **D**). (**E–**, **F**) Molecular brightness distributions based on equilibrium membrane binding of either mNG-PIP5KB or mNG-PIP5KB (D51R). Molecules with a threshold particle intensity ≥8200 (A.U.) were considered dimeric. Values printed above double arrows equal percentage of apparent dimers (N = 11,355 mNG-PIP5KB particles in **E**; N = 12,759 mNG-PIP5KB (D51R) particles in **F**). (**G**) Image showing the localization of membrane-bound mNG-PIP5KB observed in the presence of a 100 pM solution concentration. Arrows indicate mNG-PIP5KB dimers. (**H**) Intensity line scan through line draw in (**G**). (**I**) Stepwise photobleaching of a membrane-bound mNG-PIP5KB dimer. (**J**) Direct visualizing of membrane-mediated dimerization in the presence of mNG-PIP5KB. (**C–J**) Data collected in the presence of 100 pM mNG-PIP5KB or mNG-PIP5KB (D51R). Membrane composition: 96% DOPC, 4% PI(4,5)P$_2$.

The online version of this article includes the following video, source data, and figure supplement(s) for figure 4:

**Source data 1.** Related to *Figure 4A*.

**Source data 2.** Related to *Figure 4C*.

**Source data 3.** Related to *Figure 4D*.

**Source data 4.** Related to *Figure 4F*.

**Source data 5.** Related to *Figure 4G*.

*Figure 4 continued on next page*

*Figure 4 continued*

**Source data 6.** Related to *Figure 4H*.

**Source data 7.** Related to *Figure 4I*.

**Figure supplement 1.** Quantification of PIP5K dimerization-based molecular brightness.

**Figure supplement 1—source data 1.** Related to *Figure 4—figure supplement 1*.

**Figure 4—video 1.** Direct visualization of membrane-mediated dimerization of mNG-PIP5KB with single -molecule resolution.
https://elifesciences.org/articles/73747/figures#fig4video1

PIP5KB dimer interface mutation, D51R, did not suffer from protein misfolding, we introduced an R254D compensatory mutation homologous to the zPIP5KA (R287D) mutation previously shown to restore the zPIP5KA dimer interface mutation, D84R (*Figure 3B*; *Hu et al., 2015*). Kinetic analysis of PIP5KB (D51R/R254D)-driven lipid phosphorylation showed that restoring the dimer interface through this orthogonal salt bridge produced a kinase with catalytic activity indistinguishable from wild-type PIP5KB (*Figure 5C and D*).

To determine whether the differences in catalytic activity of the dimer mutant were caused by a reduction in PIP5KB (D51R) membrane recruitment or dimerization-induced change in catalytic efficiency, we compared the relative membrane binding strength of mNG-PIP5KB and mNG-PIP5KB (D51R). Over a broad range of solution concentrations, we observed approximately twofold higher membrane density for mNG-PIP5KB, compared to mNG-PIP5KB (D51R) (*Figure 5F*). This suggests that the reduction in membrane binding affinity is not the primary reason for the sevenfold reduction in activity observed when we disrupt the dimer interface of PIP5K.

Next, we simultaneously measured the kinetics of lipid phosphorylation and quantified the absolute density of membrane bound mNG-PIP5K. To determine whether dimerization enhances the catalytic efficiency of PIP5K, we calculated the effective phosphorylation rate constants per enzyme based on the calibrated surface density of mNG-PIP5KB (*Figure 5—figure supplement 2*). Compared to typical solution Michaelis–Menten kinetics, the density of membrane-bound mNG-PIP5KB changes over the course of the lipid phosphorylation reactions described in *Figure 5E*. To account for this change, the effective phosphorylation rate per membrane-bound PIP5KB molecule, $v_{molecule}(t)$, was calculated using the following equation:

$$v_{molecule}(t) = \frac{dP(t)}{dt}\frac{1}{E(t)}$$

where $E$ is the surface density of mNG-PIP5KB, and $P$ is the surface density of PI(4,5)P$_2$ on the membrane at any time during the reaction. The PI(4)P density at each time point, $S(t)$, was approximated by subtracting $P(t)$ from the initial PIP lipid density, $S_0$:

$$S(t) = S_0 - P(t)$$

By plotting $v_{molecule}(t)$ against the substrate density, $S(t)$, we obtained a Michaelis–Menten plot with a slope equal to the effective phosphorylation rate constant per kinase (*Figure 5G*). The calculated phosphorylation rate constant for the dimer mutant, PIP5KB (D51R), had a single rate constant of $2.3 \times 10^{-4}$ lipids/μm$^2$•s per enzyme throughout the observed reaction trajectory. Conversely, the phosphorylation rate constant for wild-type PIP5KB began with a slow rate and then transitioned to a rate of $2.7 \times 10^{-3}$ lipids/μm$^2$•s per kinase as the reaction progressed (*Figure 5G*, see arrows indicating the transition from slow to fast kinetics for wild-type PIP5KB). After the reaction reached the maximum velocity, the kinetics gradually slowed down due to substrate depletion. Overall, the difference in per-molecule reaction kinetics comparing PIP5KB and PIP5KB (D51R) was consistent with dimerization enhancing lipid kinase activity (*Figure 5E*) and establishes a positive feedback mechanism.

Comparing the shape of the kinetic traces for PIP5KB and PIP5KB (D51R) also revealed striking differences in the complexity of their positive feedback loops. To analyze the feedback profiles, we plotted the rate of PI(4,5)P$_2$ production, $dx/dt$, as a function of the reaction coordinate, $x$, as $x \equiv \sigma PI(4,5)P2/(\sigma PI(4,5)P2 + \sigma PI(4)P)$. $\sigma$ denotes the membrane density of each PIP lipid species throughout the entire reaction trajectory. The overall reaction rate can be expressed as

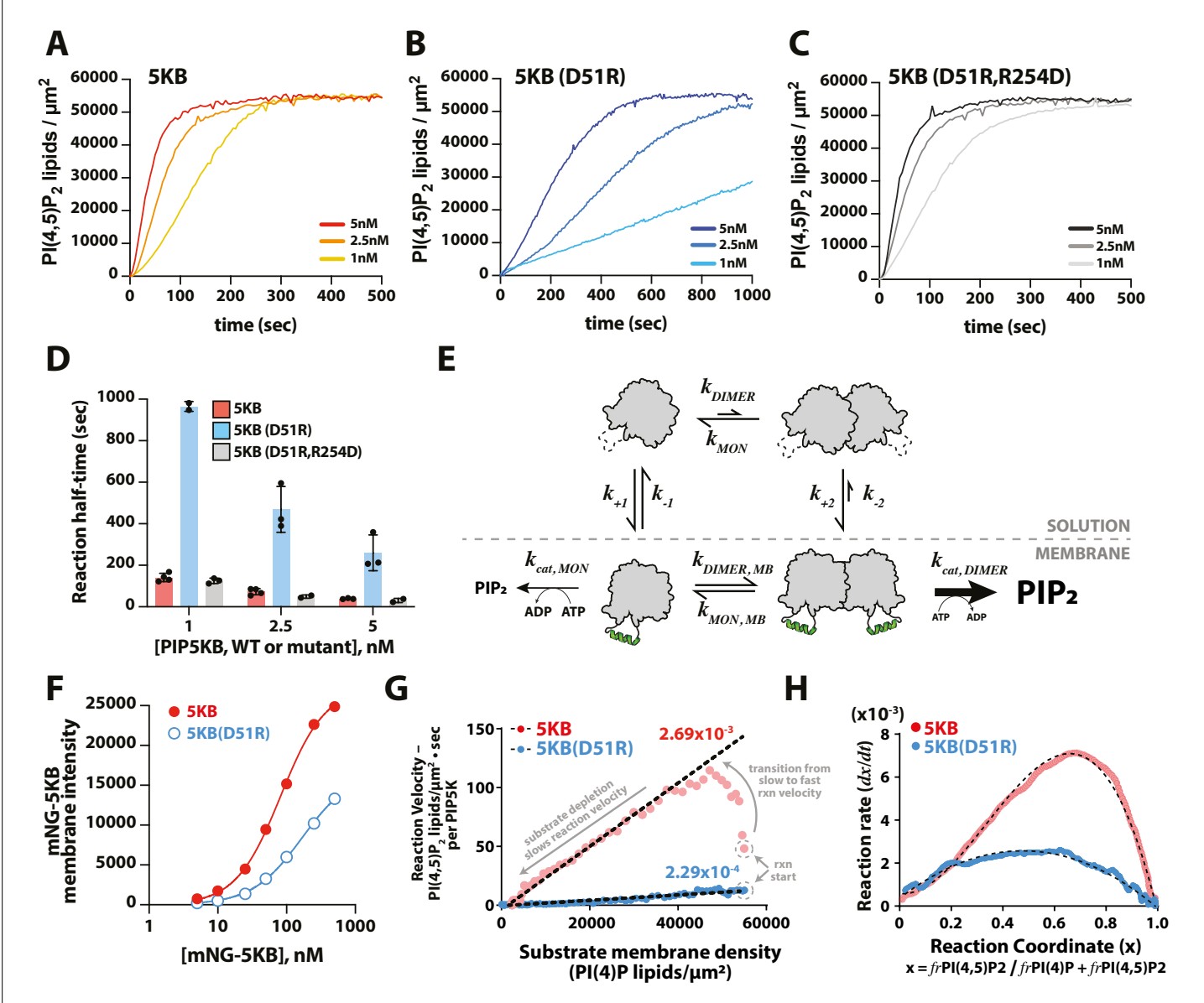

**Figure 5.** Membrane-mediated dimerization potentiates PIP5K lipid kinase activity. (**A–C**) Representative kinetic traces monitoring the production of PI(4,5)P$_2$ in the presence of 1–5 nM PIP5KB, PIP5KB (D51R), or PIP5KB (D51R/R254D). The PI(4,5)P$_2$ membrane surface density was estimated based on the membrane localization of 20 nM Ax488-PLCδ. Initial membrane composition: 96% DOPC, 4% PI(4)P. (**D**) Quantification of reaction half-time from trajectories shown in (**A–C**). Bars equal mean values (N = 2–4 reactions per concentration, error = SD). (**E**) Equilibrium diagram showing the relationship between PIP5KB membrane binding, oligomerization, and catalysis. (**F**) Equilibrium fluorescence intensity of membrane-bound mNG-PIP5KB and mNG-PIP5KB (D51R) measured at varying solution concentrations. Points are mean values (N = 15–20 fluorescent intensity measurements per sample from one experiment). Membrane composition: 96% DOPC, 4% PI(4,5)P$_2$. (**G**) Membrane-mediated dimerization enhances PIP5KB catalytic efficiency independent of membrane localization. Membrane localization of mNG-PIP5KB (WT and D51R) and production of PI(4,5)P$_2$ were simultaneously monitored on supported membranes. Each data point in the plot represents the instantaneous velocity expressed as the number of PI(4,5)P$_2$ lipids generated per μm$^2$ per second per kinase as a function of the substrate density. The reaction catalyzed by PIP5KB displays the following three phases: (1) slow kinetics at reaction start, (2) rise to maximum reaction velocity, and (3) gradual decline in reaction velocity due to substrate depletion. (**H**) Feedback profiles for PIP5KB (WT and D51R). The following equations were used for curve fitting: $\left(k_0^+ + k_1^+ x + k_2^+ x^2\right)(1-x)$ for PIP5KB and $\left(k_0^+ + k_1^+ x\right)(1-x)$ for PIP5KB (D51R) where $x$ represents the phosphatidylinositol phosphate (PIP) lipid composition.

The online version of this article includes the following source data and figure supplement(s) for figure 5:

**Source data 1.** Related to *Figure 5A*.

**Source data 2.** Related to *Figure 5B*.

*Figure 5 continued on next page*

*Figure 5 continued*

**Source data 3.** Related to *Figure 5C*.

**Source data 4.** Related to *Figure 5D*.

**Source data 5.** Related to *Figure 5F*.

**Source data 6.** Related to *Figure 5G*.

**Source data 7.** Related to *Figure 5H*.

**Figure supplement 1.** Dimerization enhances PIP5KA lipid kinase activity.

**Figure supplement 1—source data 1.** Related to *Figure 5—figure supplement 1*.

**Figure supplement 2.** Fluorescence correlation spectroscopy membrane density calibration.

**Figure supplement 2—source data 1.** Related to *Figure 5—figure supplement 2*.

$$\frac{dx}{dt} = k\left(x\right) \bullet \left(1 - x\right)$$

where $k\left(x\right)$ is a function that characterizes the reaction rate, including any dependence on the instantaneous PIP lipid composition ($x$). As previously described by *Hansen et al., 2019*, the overall rate constant, $k\left(x\right)$, can be expressed as a power series, $k\left(x\right) = k_0 + k_1 x + k_2 x^2 + \cdots$. This provides a convenient way of examining the type of feedback; the order of feedback is revealed by the $x$-dependence of $k\left(x\right)$. Plotting the derivative of the PIP5KB kinase reaction traces against the reaction coordinate (i.e., PIP lipid composition), $x$, the curve displayed a high degree of asymmetry that required a second-order term to fit ($k_{WT}\left(x\right) = k_0 + k_1 x + k_2 x^2$) (*Figure 5H*). In contrast, the $dx/dt$ curve for the PIP5KB (D51R) dimer mutant was parabolic and could be fit using an equation that describes an enzyme with simple first-order positive feedback ($k_{D51R}\left(x\right) = k_0 + k_1 x$) based on product binding (*Figure 5H*). In previous studies, PIP5KB was shown to exhibit higher-order positive feedback (*Hansen et al., 2019*). One potential source of higher-order positive feedback is through the ability of PIP5K to bind to multiple PI(4,5)P$_2$ lipids. However, mapping the feedback strength based on the reaction coordinate revealed that dimerization is predominantly responsible for PIP5K higher-order positive feedback. Together, these results provide strong evidence that dimerization enhances the activity of PIP5K by both enhancing the membrane avidity and potentiating lipid kinase activity through a mechanism that is consistent with allosteric regulation.

## PIP5K dimerization increases the stability of PIP compositional patterns

The PIP5KB positive feedback mechanism previously enabled us to reconstitute a bistable lipid kinase–phosphatase competitive reaction that breaks symmetry and produces PI(4)P and PI(4,5)P$_2$ compositional patterns on supported membranes (*Figure 6A*; *Hansen et al., 2019*). Based on the ability of PIP5KB to undergo membrane-mediated dimerization and the corresponding nonlinear positive feedback, we hypothesized the kinetic bistability in this system could be dependent on membrane-mediated dimerization. To test this hypothesis, we compared the ability of PIP5KB and the PIP5KB (D51R) dimer interface mutant to form bistable PIP compositional patterns in the presence of an opposing 5-phosphatase with engineering positive feedback based. This was achieved by fusing a minimal PI(4)P binding motif, referred to as DrrA (or SidM/DrrA), to OCRL (*Hammond et al., 2014*; *Zhu et al., 2010*). The resulting chimeric 5-phosphatase, DrrA-OCRL, was previously shown to exhibit positive feedback based on PI(4)P recognition (*Hansen et al., 2019*). In the presence of 50 nM PIP5KB (WT or D51R) and 30 nM DrrA-OCRL, we found that the activity of the D51R dimer mutant was strongly impaired compared to wild-type PIP5KB (*Figure 6B*). Restoring the PIP5KB dimer interface with a charge reversal mutation, D51R/R254D, allowed us to reconstitute bistable compositional patterns that were indistinguishable from those formed in the presence of wild-type PIP5K (*Figure 6B*). Again, the behavior of PIP5KB (D51R/R254D) indicated that the D51R mutation does not structurally harm the kinase. Taking into consideration the weakened positive feedback of the dimer mutant, we raised the solution concentration of PIP5KB (D51R) until we were able to balance the opposing 5-phosphatase activity. Using a 20-fold higher concentration of PIP5KB (D51R), compared to wild-type, we identified a concentration regime that allowed us to reconstitute PIP compositional pattern formation on supported membranes (*Figure 6B*). During the early stages of pattern formation, the surface area and morphology looked very similar to the compositional patterns reconstituted

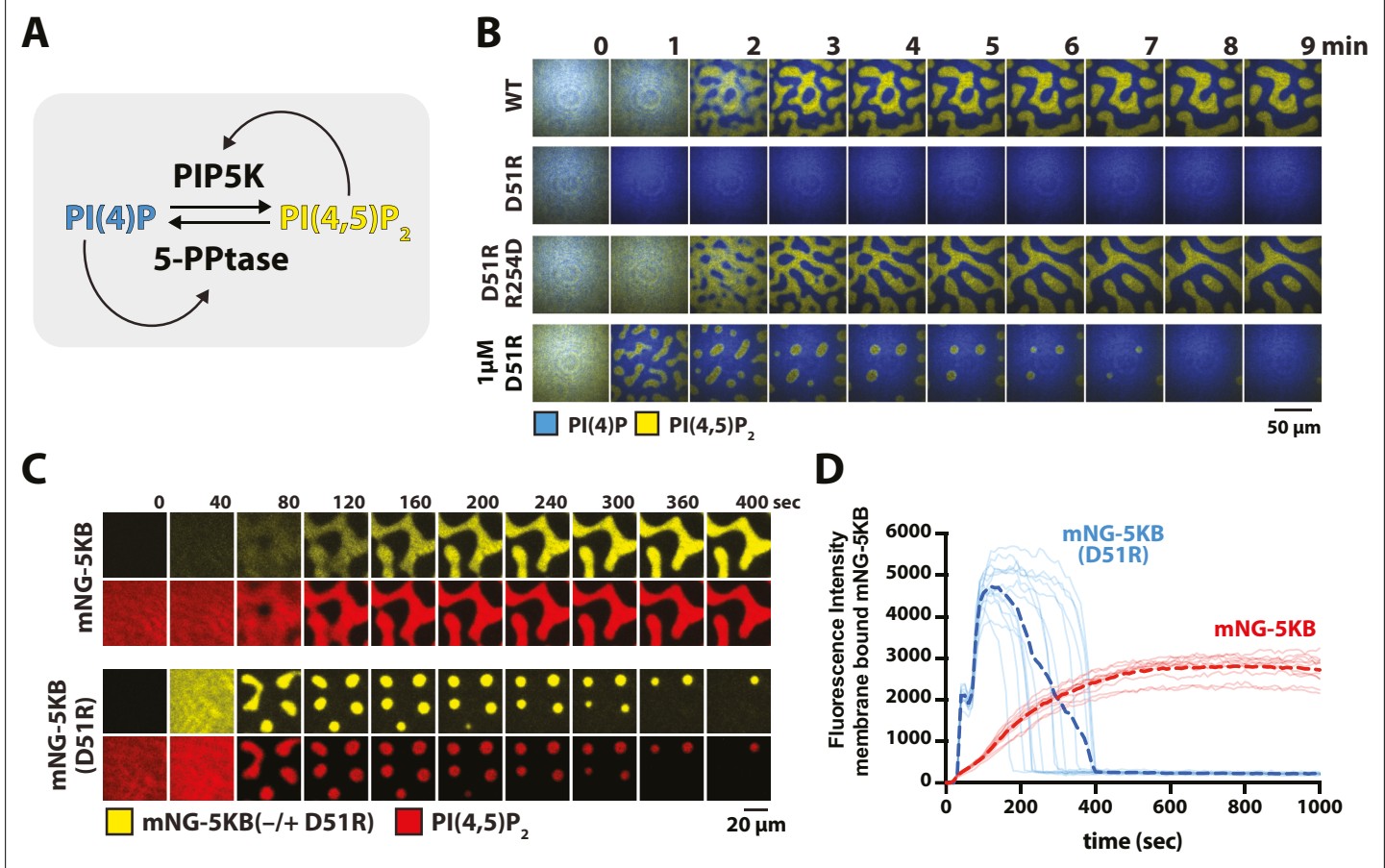

**Figure 6.** PIP5K dimerization stabilizes phosphatidylinositol phosphate (PIP) compositional patterns. (**A**) Diagram showing the network architecture of the bistable kinase–phosphatase-competitive reaction. The positive feedback loop of the 5-phosphatase, DrrA-OCRL, is regulated by product binding (*Hansen et al., 2019*). Note that DrrA is also commonly known as SidM/DrrA in *L. pneomophila*. (**B**) PIP5KB dimerization increases robustness of PIP compositional pattern formation. Montage image sequences showing the formation PIP compositional patterns in the presence 50 nM PIP5KB (WT, D51R, or D51R/R254D, montage rows 1–3) or 1 µM PIP5KB (D51R, montage row 4). (**C**) Localization of mNG-PIP5KB (WT or D51R) during PIP compositional pattern formation. Reactions were reconstitution in the presence of either 50 nM mNG-PIP5KB or 1 µM mNG-PIP5KB (D51R). (**D**) Plot showing the change in mNG-PIP5KB membrane intensity compositional patterns enriched in PI(4,5)P$_2$ lipids. The dashed line represents the mean fluorescence intensity from N = 10 membrane domains. (**B–D**) All reactions contain the specified concentration of PIP5KB (WT or mutants), plus 20 nM DrrA-OCRL, 20 nM Ax488-PLCδ, and 20 nM Ax647-DrrA. Initial membrane composition: 96% DOPC, 2% PI(4)P, 2% PI(4,5)P$_2$.

The online version of this article includes the following video and source data for figure 6:

**Source data 1.** Related to *Figure 6D*.

**Figure 6—video 1.** Visualization of mNG-PIP5KB localization during phosphatidylinositol phosphate (PIP) compositional pattern formation.
https://elifesciences.org/articles/73747/figures#fig6video1

**Figure 6—video 2.** Visualization of mNG-PIP5KB (D51R) localization during phosphatidylinositol phosphate (PIP) compositional pattern formation.
https://elifesciences.org/articles/73747/figures#fig6video2

in the presence of wild-type PIP5KB. However, after a couple of minutes the PI(4,5)P$_2$ compositional patterns generated by PIP5KB (D51R) were consumed by the surrounding 5-phosphatase dominated reaction (*Figure 6B*).

To determine whether the inability of monomeric PIP5KB (D51R) to form stable compositional patterns was due to a relatively lower membrane surface density of the dimer mutant, we compared the membrane localization dynamics of mNG-PIP5KB and mNG-PIP5K (D51R) during the formation of PIP compositional patterns (*Figure 6C*). In the presence of mNG-PIP5KB, the kinase membrane density steadily increased and then plateaued shortly after pattern formation (*Figure 6D*, *Figure 6—video 1*). In contrast, the membrane surface density of mNG-PIP5K (D51R) sharply increased immediately after initiating the kinase–phosphatase competitive reactions, reaching a density nearly two times greater

compared to the steady-state density observed for mNG-PIP5KB (*Figure 6D*, *Figure 6—video 2*). After the mNG-PIP5KB (D51R)-driven reaction reached a peak surface density that was sufficient to generate the bistable PIP compositional patterns, the PI(4,5)P$_2$-containing membrane domains rapidly decreased in size and disappeared (*Figure 6D*). Overall, mNG-PIP5K (D51R) reached a sufficiently high membrane density to form compositional patterns, but lacked the catalytic activity needed to oppose the competing 5-phosphatase reaction and maintain the established PI(4,5)P$_2$ compositional pattern.

## Reaction trajectory variation is enhanced by membrane-mediated dimerization

We previously reported that PIP5K-dependent lipid phosphorylation reactions exhibit a high degree of reaction trajectory variation when reconstituted on SLBs that are partitioned into micron-length-scale membrane corrals (*Hansen et al., 2019*). Based on the coupling induced by membrane-mediated dimerization, we hypothesized that the dimerization could provide the molecular basis for the previously observed enhanced reaction trajectory variation. To measure dimerization-dependent differences in reaction trajectory variation, we microfabricated an array of 5 µm × 5 µm chromium barrier onto the underlying glass coverslip. This approach allowed us to visualize hundreds of identical membrane reactions in parallel that continuously exchange with the surrounding solution environment. Under these conditions, PIP5KB reaction trajectories displayed a high degree of kinetic heterogeneity (*Figure 7A and B*, *Figure 7—video 1*). While the time to finish was highly heterogeneous, the reaction rate of the initial and later part of the reaction was quite homogeneous. In contrast, reactions reconstituted in the presence of PIP5KB (D51R) showed little heterogeneity, confirming that dimerization strongly enhanced variation in corral reaction trajectory (*Figure 7B*, *Figure 7—video 2*). Overall, the observed reaction heterogeneity was primarily driven at higher product concentrations, where higher-order positive feedback dominates. Formation of the PIP5K dimer creates a highly active kinase with a strengthened positive feedback and faster reaction velocity.

## Stochastic effects can enhance bistability

Nonlinear positive feedback is sufficient to establish bistability in a competitive reaction, even on scales over which stochastic variations average to essentially zero. The stable patterns seen in *Figure 6B* (wild-type and D51R/R254D) are likely a representation of this non-stochastic bistability, with two intrinsically stable steady states. The dimer mutant, however, shows only linear positive feedback (see *Figure 5H*) but still shows at least transient bistability in *Figure 6B–D*. This is a manifestation of stochastic bistability, referring to bistable behavior in systems that inherently lack two stable steady states (*Bishop and Qian, 2010*; *Hansen et al., 2019*; *Artyomov et al., 2007*; *To and Maheshri, 2010*). For the wild-type PIP5KB, dimerization enhances stochastic variation in reaction rate and thus we hypothesize will expand the range of conditions over which bistable behavior is possible, even spanning beyond the boundaries of intrinsic bistability. To test this hypothesis, we examine competitive reactions with OCRL phosphatase for PIP5KB and PIP5KB (D51R) under geometric confinement, where stochastic effects are prominent. Reconstitution of the PIP5KB and OCRL competitive reactions in 5 µm × 5 µm membrane corrals reveals strong bistability, reaching final membrane compositions in each corral consisting of steady states dominated by either PI(4)P or PI(4,5)P$_2$ composition (*Figure 7C*). To compare with the dimer mutant, we titrated the competing OCRL concentration against fixed concentrations of either PIP5KB and PIP5KB (D51R) and examine the resultant bistability (*Figure 7D*). Wild-type PIP5KB exhibits a much more robust bistability, which spans a substantially wider range of competing phosphatase concentrations than observed for the PIP5KB (D51R) dimer mutant. Quantification of this effect is plotted in *Figure 7E*. Dimerization of PIP5K thus ensures that the kinase–phosphatase-competitive reactions can achieve a bistable response over a broad range of opposing 5-phosphatase activity.

## Discussion
### Dimerization as a mechanism for potentiating PIP5K activity

Cooperative PI(4,5)P$_2$ binding and membrane-mediated dimerization provide synergistic mechanisms to increase the rate of PI(4,5)P$_2$ production through the enhanced localization and increased catalytic

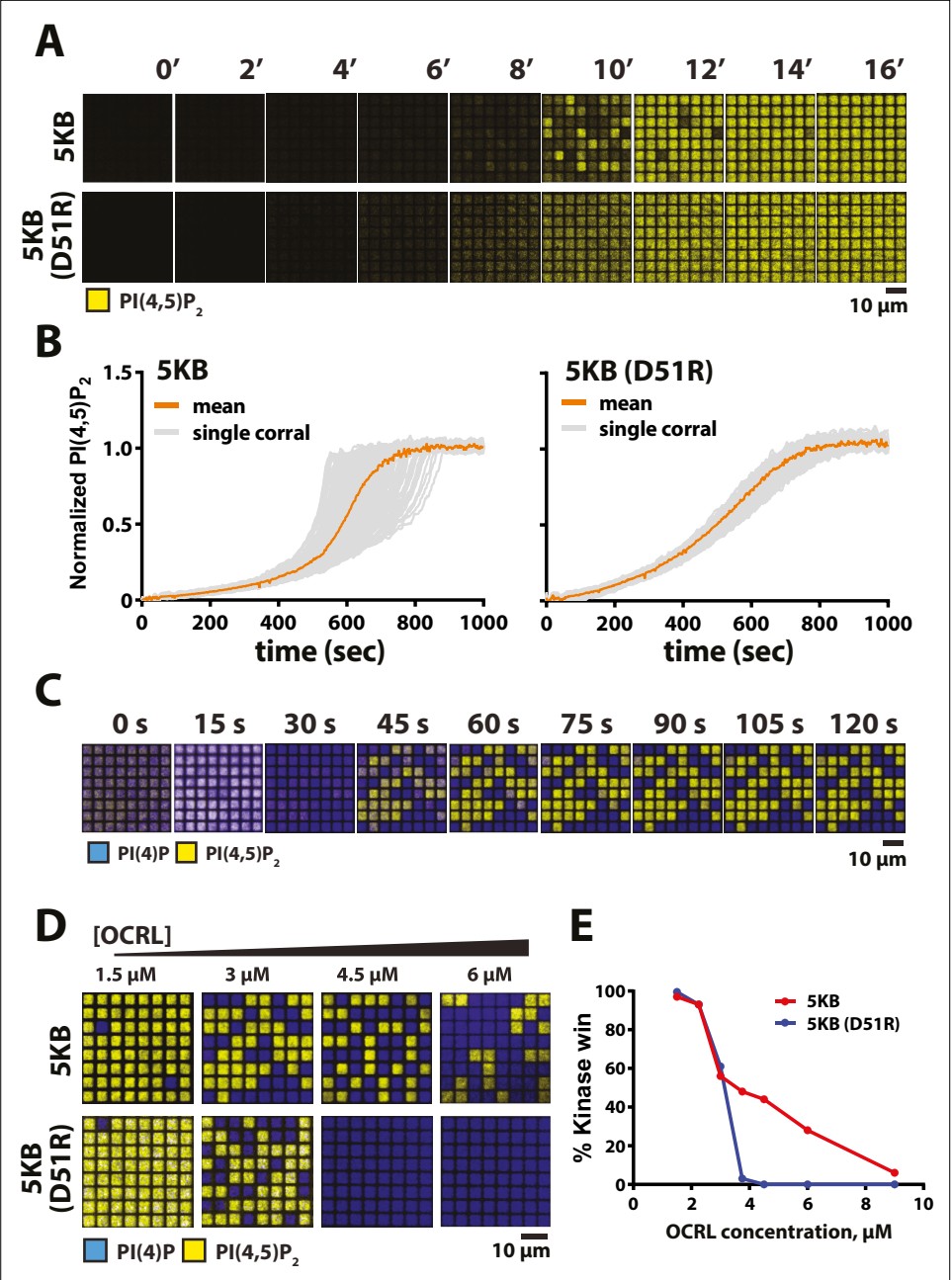

**Figure 7.** Reaction trajectory variation based on stochastic membrane-mediated dimerization. (**A**) Lipid phosphorylation reactions reconstituted in 5 μm × 5 μm chromium-patterned-supported membranes in the presence of 5 nM PIP5K and 10 nM PIP5K (D51R). Reactions were visualized using 20 nM Alexa488-PLCδ. Initial membrane composition: 96% DOPC, 4% PI(4)P. (**B**) Reaction trajectories plots from (**A**) from N = 100 membrane corrals. The mean reaction trajectory is plotted in orange. (**C**) Time course of bistable kinase–phosphatase reaction reconstituted in 5 μm × 5 μm corrals in the presence of 50 nM PIP5K, 30 nM DrrA-OCRL, 20 nM Alexa488-PLCδ, and 20 nM Ax647-DrrA. (**D**) Dimerization enhances the ability of PIP5K to win competitive bistable kinase–phosphatase reaction in the presence of OCRL. Representative steady-state images showing outcomes reconstituted in the presence of 3 μM OCRL, plus either 18 nM PIP5KB or 370 nM PIP5KB (D51R). The concentration of PIP5KB (WT and D51R) used for these experiments was determined based on achieving an approximately 50–50 reaction outcome in the presence of 3 μM OCRL. (**E**) Quantification of final reaction outcome in (**D**) from a single experiment (N > 100 corrals per OCRL concentration). (**C–E**) Initial membrane composition: 96% DOPC, 2% PI(4)P, and 2% PI(4,5)P$_2$.

The online version of this article includes the following video and source data for figure 7:

*Figure 7 continued on next page*

*Figure 7 continued*

**Source data 1.** Related to *Figure 7B*.

**Source data 2.** Related to *Figure 7C*.

**Source data 3.** Related to *Figure 7E*.

**Figure 7—video 1.** Visualization of lipid phosphorylation reactions reconstituted in 5 µm x 5 µm chromium -patterned -supported membranes.

https://elifesciences.org/articles/73747/figures#fig7video1

**Figure 7—video 2.** Visualization of lipid phosphorylation reactions reconstituted in 5 µm x 5 µm chromium -patterned -supported membranes.

https://elifesciences.org/articles/73747/figures#fig7video2

efficiency of PIP5K (*Figure 8*). Supporting a mechanism of allosteric regulation, our results indicate that dimerization enhances PIP5K lipid kinase activity by directly increasing $k_{cat}$. This effect is independent of increasing membrane avidity of PIP5K. Due to a lack of structural biochemistry, there is currently a gap in knowledge concerning the role dimerization, PIP lipid binding, and the nucleotide state serve in regulating conformational states of PIP5K. Given our limited structural understanding of PIP5K (*Liu et al., 2016*; *Hu et al., 2015*; *Muftuoglu et al., 2016*), some researchers have used molecular dynamic simulations to elucidate how membrane docking of PIP5K is controlled (*Amos et al., 2019*). Working under the assumption that PIP5K is constitutively dimeric, Amos et al. reported that only a single kinase domain can engage substrate when the dimer is docked on a PI(4)P-containing membrane. Consistent with this observation, we find that the dimerization modestly enhanced membrane binding. If this mechanism is accurate, membrane-mediated dimerization is expected to enhance processivity of membrane-bound PIP5K by allowing kinase domains to toggle between states of catalysis and membrane binding. This mechanism would ensure that dimeric PIP5K remains bound to the membrane during enzyme catalysis, while the monomeric PIP5K would likely catalyze a single PI(4)P phosphorylation reaction before dissociating. According to our single-molecule biophysical studies, we believe that both subunits of the PIP5K kinase domain can engage the membrane. This is based on the enhanced dwell time and slower diffusion coefficient observed when Ax647-PIP5K is

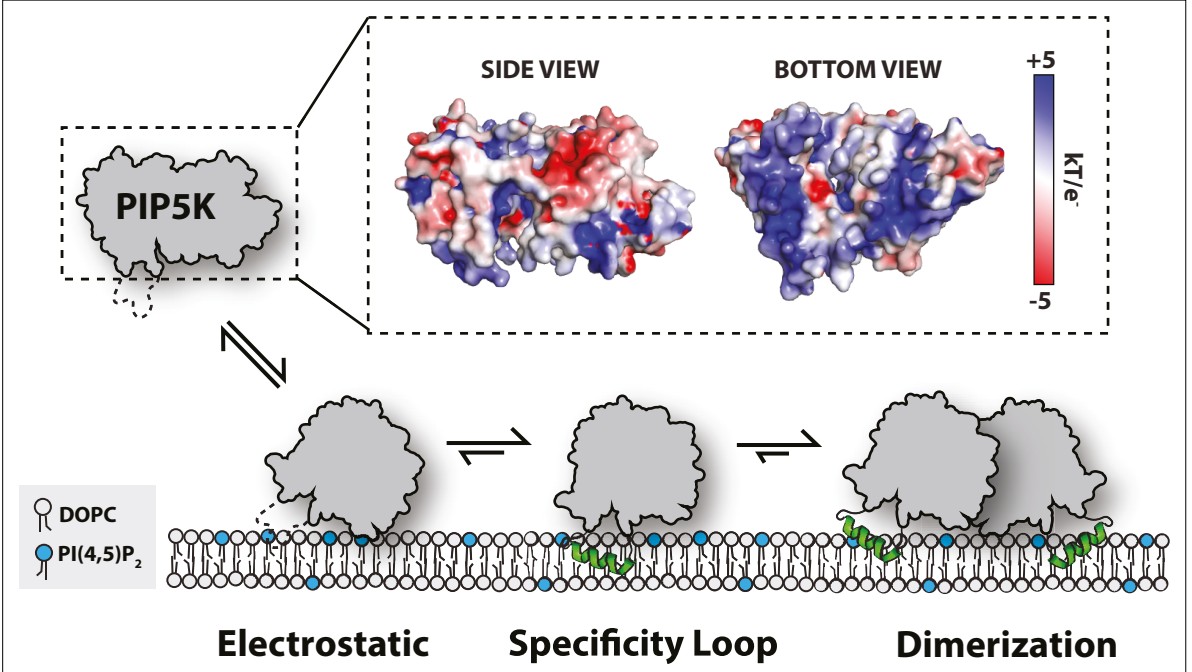

**Figure 8.** Model for PIP5K membrane-mediated dimerization. Mediated by electrostatic interactions, PIP5K can associate with membranes as a monomer. The specificity loop stabilizes membrane association and enables PIP5K to catalyze the phosphorylation of PI(4)P to generate PI(4,5)P$_2$. Increasing densities of PI(4,5)P$_2$ and membrane-bound PIP5K promote membrane-mediated dimerization, which leads to enhanced catalytic efficiency.

visualized is high protein density (i.e., ~100 molecules/$\mu m^2$). Incorporating reversible dimerization, catalysis, and PI(4,5)$P_2$ binding into future molecular dynamic simulations could provide new insight into the mechanism of membrane docking.

## PIP5K dimerization increases reaction trajectory variation

Cooperative PI(4,5)$P_2$ binding, membrane-mediated dimerization, and low molecular copy number of PIP5K provide several mechanisms for enhancing stochasticity in membrane-associated lipid phosphorylation reactions. Previously, we reported that reaction velocity fluctuations driven by stochastic binding and unbinding of PIP5K on the membrane could drive the system to a predominantly PI(4,5)$P_2$ state, even in the presence of high levels of opposing lipid phosphatase activity that were sufficient to drive the system predominantly to the PI(4)P state in bulk (*Hansen et al., 2019*). The resulting emergent property is that the reaction outcome depends on system size, which we termed stochastic geometry sensing; related scale-dependent phenomena have been called reaction inversion in some theoretical work (*Ramaswamy et al., 2012*). Here, we report that dimerization is responsible for the higher-order positive feedback exhibited by PIP5K, and this contributes to the robustness with which this system exhibits stochastic geometry sensing. Without dimerization, PIP5K displays a simple linear positive feedback profile. Comparing the lipid phosphorylation reaction trajectory variation of PIP5KB and PIP5KB (D51R) highlights how dimerization enhances stochasticity of lipid kinase phosphorylation reactions and will thus amplify any effects dependent on stochastic fluctuations. Given the broad dynamic range of kinase densities and activities that can emerge from these biochemical mechanisms, stochasticity in PIP5K signaling has the potential to strongly influence signaling events orchestrated at the plasma membrane in living cells.

## PIP5K localization and lipid kinase activity in cells

The strong lipid phosphorylation activity of PIP5K observed in vitro raises questions about the steady-state localization and activity of PIP5K in cells. Although new molecular mechanisms concerning PIP5K activation have been revealed through single-molecule characterization of PIP5K in vitro, it remains challenging to interpret how dimerization, PI(4,5)$P_2$ binding, and interactions with peripheral membrane proteins regulate membrane localization of PIP5K in vivo. Complicating our interpretation of cellular localization, PIP5K can also reportedly interact with phosphatidylserine and sterol lipids, which modulate lipid kinase activity (*Nishimura et al., 2019*). In addition, PIP5KA, PIP5KB, and PIP5KC paralogs can potentially form heterodimeric complexes that have unique membrane avidities and kinase activities (*Lacalle et al., 2007*).

Human proteomic data has estimated the cellular concentration of PIP5K in mammalian cells to be 10–20 nM (*Hein et al., 2015*). Based on the weak monomer–dimer equilibrium, PIP5K is predicted to exist predominantly as a monomer in the cytoplasm. However, receptor activation could shift the steady-state localization of PIP5K to favor membrane-dependent dimerization and enhanced lipid kinase activity. This could be achieved by increasing PIP5K membrane recruitment through interactions with receptors, endocytic machinery, or small GTPases (*Halstead et al., 2010*; *Honda et al., 1999*; *Funakoshi et al., 2010*). If the steady-state localization of PIP5K exists near a threshold for activation, a minor enhancement in localization could promote PIP5K dimerization and trigger the PI(4,5)$P_2$-dependent positive feedback loop. Future biochemical characterization of PIP5K will enable researchers to generate better separation of function mutants that allow the functional significance of specific protein–protein and protein–lipid interactions to be studied in the context of cell signaling. Determining how cells regulate the strength and duration of PIP5K lipid phosphorylation reactions during receptor activation will also help elucidate the role cooperative PI(4,5)$P_2$ binding and dimerization serve in enhancing PIP5K-dependent production of PI(4,5)$P_2$ during cell signaling.

One challenge in establishing an accurate model for PIP5K membrane binding and catalysis is the gap in knowledge concerning how PIP5K binds to PI(4,5)$P_2$ and other lipids. Previously published cell biology data reported that several basic residues are critical for PIP binding and plasma membrane localization in cells (*Fairn et al., 2009*). However, these basic residues are dispensable for PIP5K membrane binding and activity in vitro (*Hansen et al., 2019*). Looking at the crystal structure of zPIP5KA (*Hu et al., 2015*; *Muftuoglu et al., 2016*), there are several motifs of unresolved electron density and domains extending beyond the N- and C-terminus of the kinase domain for which we lack high-resolution structural data. Defining the functional relevance of those domains in controlling

PIP5K membrane association could provide new insight into the molecular basis of PI(4,5)P$_2$-binding specificity. Considering the cooperative nature of the PIP5K and PI(4,5)P$_2$ interaction, visualization of fluorescently labeled PIP5K in cells could be used to infer local membrane density of PI(4,5)P$_2$ at the plasma membrane based on single-molecule dwell times of PIP5K. We know from previous work that PIP5K becomes enriched at synaptic vesicles (*Nakano-Kobayashi et al., 2007*) and focal adhesions (*Ling et al., 2002*; *Di Paolo et al., 2002*). However, it remains unclear whether localization of PIP5K to these cellular structure requires interactions with PI(4,5)P$_2$ lipids, dimerization of the kinase domain, or recruitment by peripheral membrane proteins.

## Negative regulation of PIP5K signaling

The strong positive feedback loop displayed by PIP5K raises questions concerning how lipid kinase activity is inhibited once production of PI(4,5)P$_2$ is accelerated. Left unregulated, the PIP5K-positive feedback loop has the potential to generate excessively high concentrations of PI(4,5)P$_2$ in cells, which would be detrimental to numerous signaling pathways that rely on cellular PIP lipid homeostasis. New evidence suggests that, in vivo and in vitro, PIP4K can attenuate PIP5K activity through the formation of a membrane-bound hetero-kinase complex (*Wang et al., 2019*; *Wills et al., 2022*). Deciphering the molecular basis of PIP4K-PIP5K complex formation using single-molecule in vitro measurements will be critical for determining both the mechanism of kinase inhibition and for generating separation of functions mutants that perturb this regulatory mechanism.

By limiting the concentration of freely available PI(4,5)P$_2$, cells can potentially control the strength and duration of the PIP5K-positive feedback loops. In vitro, PIP5K interacts strongly with supported membranes containing 2–4% PI(4,5)P$_2$. Although the total concentration of PI(4,5)P$_2$ is estimated to be 0.5–5% in the plasma membrane (*Wenk et al., 2003*; *Mitchell et al., 1986*; *Nasuhoglu et al., 2002*), the concentration of freely available PI(4,5)P$_2$ is potentially lower. PI(4,5)P$_2$-sequestering proteins, like MARCKS, are present at micromolar concentrations in cells (*Brudvig and Weimer, 2015*) and likely limit the concentration of freely available PI(4,5)P$_2$ at the plasma membrane. Lipid phosphatases, phosphatidylinositol-3 kinase (PI3K), and phospholipases also regulate the conversion of PI(4,5)P$_2$ into other lipid species and secondary metabolites (*Balla, 2013*). In the case of early phagocytosis, macrophages can put the brakes on PIP5K activity by activating phospholipase C, which cleaves the inositol head group from PI(4,5)P$_2$ to produce second messenger, IP3 and DAG (*Rhee, 2001*). Parallel to this pathway, PI3K phosphorylates PI(4,5)P$_2$ to generate PI(3,4,5)P$_3$. Both enzymatic reactions reduce the amplitude and duration of PI(4,5)P$_2$ spikes. By understanding how lipid kinases, phosphatases, and sequestering molecules indirectly regulate the activity of PIP5K through modulation of membrane lipid composition, we will gain a deeper understanding of how cells control PI(4,5)P$_2$ lipid homeostasis. Considering the myriad of human diseases linked to the disruption of PIP lipid homeostasis, there is much to gain by understanding the molecular mechanisms that regulate the spatiotemporal dynamics of PIP5K lipid kinase activity in cells.

## Materials and methods
### Molecular biology

The gene coding for *Legionella pneomophila* DrrA/SidM (accession #Q5ZSQ3.1) was used to clone the minimal PI(4)P lipid-binding domain, which is referred to as DrrA throughout the article and *Hansen et al., 2019* . Human phosphatidylinositol 4,5-bisphosphate phosphodiesterase delta-1 PH domain (PLCδ, accession #P51178.2), human oculocerebrorenal syndrome of Lowe inositol polyphosphate 5-phosphatase (OCRL; UniProt #Q01968), human phosphatidylinositol 5-phosphate 4-kinase type-2 beta (PIP4KB; UniProt #P78356), and human phosphatidylinositol 4-phosphate 5-kinase type-1 beta isoform 2 (PIP5KB; UniProt #O14986) were derived from codon-optimized genes synthesized by GeneArt (Invitrogen). The gene encoding yeast Mss4 (UniProt #P38994) was obtained by PCR amplification from yeast genomic DNA (S288C, *Saccharomyces cerevisiae*). The gene encoding mouse PIP5KA isoform 1 was purchased as a cDNA clone from Horizon Discovery (Cat# MMM1013-202762630, UniProt #P70182). The gene encoding human PIP5KC1 (UniProt #O60331) was provided by Peter Bieling (Max Planck Institute of Molecular Physiology, Dortmund, Germany). The plasmids containing codon-optimized zebrafish PIP5KA (zPIP5KA 49-431aa, UniProt #A0A2R8QMJ9, accession # NP_001018438.1) for expression in bacteria were kindly provided by Jian Hu (Michigan State

University, Department of Biochemistry and Molecular Biology). Gene sequences were subcloned into either bacterial, insect cell, or mammalian expression vectors using ligation-independent cloning or Gibson assembly (*Gibson et al., 2009*). The open-reading frame of all vectors used in this study was sequenced to ensure that protein expression constructs lacked deleterious mutations.

Note that there is some confusion in the literature concerning the naming of mouse and human PIP5K homologs. For example, some articles refer to the human α form (i.e., hPIP5KA) as the mouse β form (i.e., mPIP5KB). This confusion appears to have been corrected in the UniProt gene and protein sequence database, thus making mouse PIP5KA (UniProt #P70182) the direct homolog of human PIP5KA (UniProt #Q99755). Similarly, mouse PIP5KB (UniProt #P70181) the direct homolog of human PIP5KB (UniProt #O14986). In this study, the sequence of mouse PIP5KA used is homologous to human PIP5KA and distinct from human PIP5KB. Refer to *Supplementary file 1* for the exact amino acid sequences used for recombinant protein expression and transient transfection in HEK293T cells.

## Purification of BACMID DNA

To create BACMID DNA, FASTBac1 plasmids were transformed into DH10 Bac cells and plated on agar containing 50 µg/mL kanamycin, 10 µg/mL tetracycline, 7 µg/mL gentamycin, 40 µg/mL X-GAL, and 40 µg/mL IPTG. After 2–3 days of growth at 37°C, positive clones were isolated based on blue-white colony selection. Single white colonies were picked and struck on a BACMID agar plate for a second round of selection. BACMIDs were purified from 3 mL bacterial cultures grown overnight in TPM. Bacteria were centrifuged and resuspended in 300 µL of buffer containing 50 mM Tris (pH 8.0), 10 mM EDTA, 100 µg/mL RNase A (QIAGEN PI buffer). Bacteria were then lysed by adding 300 µL of buffer containing 200 mM NaOH, 1% SDS (QIAGEN P2 buffer). Lysis buffer was neutralized by adding 300 µL of 4.2 M guanidine HCl, 0.9 M KOAc (pH 4.8) (QIAGEN N3 buffer). Sample was centrifuged at 23°C for 10 min at 14,000 × $g$. Supernatant was removed and combined with 700 µL 100% isopropanol. Sample was centrifuged at 23°C for 10 min at 14,000 × $g$. Supernatant was removed and 200 µL of 70% ethanol was added. Sample was centrifuged at 23°C for 10 min at 14,000 × $g$. Supernatant was removed and 50 µL of 70% ethanol was added. Sample was centrifuged at 23°C for 10 min at 14,000 × $g$. Supernatant was removed and DNA pellet was slightly dried in biosafety hood. DNA was solubilized with 40 µL of sterile water. DNA pellet was resuspended by tapping side of micro-centrifuge tube 15–20 times. Concentration of DNA was quantified using NanoDrop (typically 200–300 ng/µL). BACMID DNA was immediately used for transfection of *Sf9* cells. This cell line was derived from the *Spodoptera frugiperda* (*Sf9*) cell line IPLB-Sf-21-AE (Expression Systems, Cat# 94-001F). The remaining BACMID DNA can be stored in –20°C freezer.

## Baculovirus production

Baculoviruses were generated by transfecting 1 × 10⁶ *S. frugiperda* (*Sf9*) insect cells plated for 24 hr in a Corning 6-well plastic dish (Cat# 07-200-80) containing 2 mL of ESF 921 Serum-Free Insect Cell Culture media (Expression Systems, Cat# 96-001, Davis, CA). All media contain 1× concentration of Antibiotic-Antimycotic (Gibco/Invitrogen, Cat# 15240-062). For transfection, 5–7 µg BACMID DNA was incubated with 4 µL Fugene (Thermo Fisher, Cat# 10362100) in 200 µL of ESF 921 media for 30 min at 23°C. BACMID DNA and Fugene were added dropwise to 6-well dish. Media change before and after addition of transfection reagent is unnecessary. After 4–5 days of transfection, viral supernatant (termed 'P0') was harvested, centrifuged, and used to infect 7 × 10⁶ *Sf9* cells plated for 24 hr in 10 cm tissue culture grade Petri dish containing 10 mL of ESF 921 media and 10% fetal bovine serum (Seradigm, Cat# 1500-500, Lot# 176B14). After 4 days, viral supernatant (termed 'P1') was harvested and centrifuged to remove cell debris. Typical P1 viral titer yield is 10–12 mL. The P1 viral titer was expanded in 100 mL *Sf9* cell culture grown to a density of 1.25–1.5 × 10⁶ cells/mL in a sterile 250 mL polycarbonate Erlenmeyer flask with vented cap (Corning, #431144). We typically transduced 100 mL *Sf9* culture with a concentration of 1% vol/vol of PI viral titer. The remaining PI virus was frozen as 1.5 mL aliquots that were stored in the –80°C freezer. The 10% fetal bovine serum served as a cryo-protectant. After 4 days, viral supernatant (termed 'P2') was harvested, centrifuged, and 0.22 µm filtered in 150 mL filter-top bottle (Corning, polyethersulfone [PES], Cat# 431153). The P2 viral supernatant is used for protein expression in High Five cells grown in ESF 921 Serum-Free Insect Cell Culture media. The multiplicity of infection (MOI) for optimal protein expression is determined

empirically to minimize cell death and maximize protein yield (typically 1.5–2% vol/vol final concentration of P2 virus).

## Protein purification

### PIP5KA, PIP5KB, PIP5KC, zPIP5KA, yMss4

Gene sequences encoding PIP5K family proteins and dimer interface mutants were cloned into FastBac1 vectors in frame with an N-terminal his6-MBP-(Asn)$_{10}$-TEV-GGGGG. Protein expression in High Five insect cells was constitutive under the polyhedrin (pH) promoter. High Five cells were infected with baculovirus using an optimized MOI, typically 1.5–2% vol/vol, was determined empirically from small-scale test expression (25–50 mL culture). Infected cells were typically grown for 48 hr at 27°C in ESF 921 Serum-Free Insect Cell Culture medium (Expression Systems, Cat# 96-001-01). Cells were harvested by centrifugation, washed with 1× PBS (pH 7.2), and then stored in the –80°C freezer. Flash freezing the cell pellet with liquid nitrogen prior to storage was not essential to purify functional kinase. For purification, frozen insect cell pellets for 4–6 L of liquid culture were thawed in an ambient water bath and lysed into buffer containing 50 mM Na$_2$HPO$_4$ (pH 8.0), 10 mM imidazole, 400 mM NaCl, 1 mM PMSF, 5 mM BME, 100 µg/mL DNase, 1 Sigma protease inhibitor cocktail EDTA free per 100 mL lysis buffer using a dounce homogenizer. Lysate was centrifuged at 35,000 rpm (140,000 × $g$) for 60 min under vacuum using a Beckman Ti-45 rotor at 4°C. Lysate was then batch bound to 5 mL of Ni-NTA Agarose (QIAGEN, Cat# 30230) resin at 4°C for 1–2 hr in a beaker set on a stir plate. Resin was then collected in 50 mL tubes, centrifuged, and washed with buffer containing 50 mM Na$_2$HPO$_4$ (pH 8.0), 10 mM imidazole, 400 mM NaCl, and 5 mM BME before being transferred to gravity flow column. NiNTA resin with his6-MBP-(Asn)$_{10}$-TEV-GGGGG-PIP5K (or yeast Mss4) was then washed with 100 mL of 50 mM Na$_2$HPO$_4$ (pH 8.0), 30 mM imidazole, 400 mM NaCl, and 5 mM BME buffer and then eluted into buffer containing 500 mM imidazole. Peak fractions were pooled, combined with 200 µg/mL his6-TEV(S291V) protease, and dialyzed against 4 L of buffer containing 20 mM Tris (pH 8.0), 200 mM NaCl, 2.5 mM BME for 16–18 hr at 4°C. Dialysate was then combined 1:1 with 20 mM Tris (pH 8.0), 1 mM DTT (~100 mM NaCl final). Precipitation was removed by centrifugation and 0.22 µm syringe filtration. Clarified dialysate was then bound to a MonoS cation-exchange column (GE Healthcare, Cat# 17-5168-01) equilibrated in 20 mM Tris (pH 8.0), 100 mM NaCl, 1 mM TCEP buffer. Proteins were resolved over a 10–100% linear gradient (0.1–1 M NaCl, 45 CV, 45 mL total, 1 mL/min flow rate). PIP5K homologs and paralogs typically eluted from the MonoS column in the presence of 370–450 mM NaCl. Peak fractions containing PIP5K (or Mss4) were pooled, concentrated in a 30 kDa MWCO Vivaspin 6 centrifuge tube (GE Healthcare, Cat# 28-9323-17), and loaded onto a 24 mL Superdex 200 10/300 GL (GE Healthcare, Cat# 17-5174-01) size-exclusion column equilibrated in 20 mM Tris (pH 8.0), 200 mM NaCl, 10% glycerol, 1 mM TCEP. Peak fractions were concentrated in a 30 kDa MWCO Vivaspin 6 centrifuge tube and snap-frozen at a final concentration of 20–40 µM using liquid nitrogen.

### PIP4K2B

Codon-optimized gene sequence encoding human PIP4K2B isoform 2 (UniProt # P78356) was cloned into a pETM-derived bacterial expression vector to create the following fusion protein: his6-SUMO3-GGGGG-PIP4K2B (1-416aa). Throughout the article, PIP4K2B is referred to as PIP4K. Recombinant PIP4K2B was expressed in BL21(DE3) Star *Escherichia coli* (i.e., lack endonuclease for increased mRNA stability). Using 2–4 L of Terrific Broth, bacterial cultures were grown at 37°C until OD$_{600}$ = 0.6. Cultures were then shifted to 18°C for 1 hr to cool down. Protein expression was induced with 50 µM IPTG and bacteria-expressed protein for 20 hr at 18°C before being harvested by centrifugation. For purification, cells were lysed into buffer containing 50 mM Na$_2$HPO$_4$ (pH 8.0), 400 mM NaCl, 0.4 mM BME, 1 mM PMSF (add twice, 15 min intervals), DNase, 1 mg/mL lysozyme using a microtip sonicator. Lysate was centrifuged at 16,000 rpm (35,172 × $g$) for 60 min in a Beckman JA-17 rotor chilled to 4°C. Lysate was circulated over 5 mL HiTrap Chelating column (GE Healthcare, Cat# 17-0409-01) that had been equilibrated with 100 mM CoCl$_2$ for 1 hr, washed with Milli-Q water, and followed by buffer containing 50 mM Na$_2$HPO$_4$ (pH 8.0), 400 mM NaCl, 0.4 mM BME. Recombinant PIP4K2B was eluted with a linear gradient of imidazole (0–500 mM, 8 CV, 40 mL total, 2 mL/min flow rate). Peak fractions were pooled, combined with 50 µg/mL of his6-SenP2 (SUMO protease), and dialyzed against 4 L of buffer containing 25 mM Na$_2$HPO$_4$ (pH 8.0), 400 mM NaCl, and 0.4 mM BME for 16–18 hr at 4°C. Following

overnight cleavage of the SUMO3 tag, dialysate-containing his6-SUMO3, his6-SenP2, and GGGGG-PIP4K2B was recirculated for at least 1 hr over a 5 mL HiTrap (Co$^{2+}$) Chelating column. Flow-through containing GGGGG-PIP4K2B was then concentrated in a 30 kDa MWCO Vivaspin 6 before loading onto a Superdex 200 size-exclusion column equilibrated in 20 mM HEPES (pH 7), 200 mM NaCl, 10% glycerol, 1 mM TCEP. In some cases, cation-exchange chromatography was used to increase the purity of GGGGG-PIP4K2B before loading on the Superdex 200. In those cases, we equilibrated a MonoS column 20 mM HEPES (pH 7), 100 mM NaCl, 1 mM TCEP buffer. PIP4K2B (pI = 6.9) bound to the MonoS was resolved over a 10–100% linear gradient (0.1–1 M NaCl, 30 CV, 30 mL total, 1.5 mL/min flow rate). Peak fractions collected from the Superdex 200 were concentrated in a 30 kDa MWCO Vivaspin 6 centrifuge tube and snap-frozen at a final concentration of 20–80 µM using liquid nitrogen.

## PLCδ-PH domain

The coding sequence of human PLCδ-PH domain (11-140aa) was expressed in BL21 (DE3) Star *E. coli* as a his$_6$-SUMO3-(Gly)$_5$-PLCd (11-140aa) fusion protein. Bacteria were grown at 37°C in Terrific Broth to an OD$_{600}$ of 0.8. Cultures were shifted to 18°C for 1 hr, induced with 0.1 mM IPTG, and allowed to express protein for 20 hr at 18°C before being harvested. Cells were lysed into 50 mM Na$_2$HPO$_4$ (pH 8.0), 300 mM NaCl, 0.4 mM BME, 1 mM PMSF, 100 µg/mL DNase using a microfluidizer. Lysate was then centrifuged at 16,000 rpm (35,172 × *g*) for 60 min in a Beckman JA-17 rotor chilled to 4°C. Lysate was circulated over 5 mL HiTrap Chelating column (GE Healthcare, Cat# 17-0409-01) charged with 100 mM CoCl$_2$ for 1 hr. Bound protein was then eluted with a linear gradient of imidazole (0–500 mM, 8 CV, 40 mL total, 2 mL/min flow rate). Peak fractions were pooled, combined with SUMO protease (50 µg/mL final concentration), and dialyzed against 4 L of buffer containing 50 mM Na$_2$HPO$_4$ (pH 8.0), 300 mM NaCl, and 0.4 mM BME for 16–18 hr at 4°C. Dialysate-containing SUMO cleaved protein was recirculated for 1 hr over a 5 mL HiTrap Chelating column. Flow-through containing (Gly)$_5$-PLCδ (11-140aa) was then concentrated in a 5 kDa MWCO Vivaspin 20 before being loaded on a Superdex 75 size-exclusion column equilibrated in 20 mM Tris (pH 8.0), 200 mM NaCl, 10% glycerol, 1 mM TCEP. Peak fractions containing (Gly)$_5$-PLCδ (11-140aa) were pooled and concentrated to a maximum concentration of 75 µM (1.2 mg/mL) before snap-freezing with liquid nitrogen and storage at –80°C.

## OCRL and DrrA

The coding sequence of human 5-phosphatase OCRL (234-539aa of 901aa isoform) was expressed in BL21 (DE3) *E. coli* as a his6-MBP-(Asn)$_{10}$-TEV-(Gly)$_5$-OCRL fusion protein. DrrA/SidM (544-647aa of 647aa gene) derived from *L. pneomophila* was expressed in BL21 (DE3) *E. coli* as a his6-MBP-(Asn)$_{10}$-TEV-(Gly)$_5$-DrrA(544-647aa) fusion protein. For the proteins described above, bacteria were grown at 37°C in Terrific Broth to an OD$_{600}$ of 0.8. Cultures were shifted to 18°C for 1 hr, induced with 0.1 mM IPTG, and allowed to express protein for 20 hr at 18°C before being harvested. Cells were lysed into 50 mM Na$_2$HPO$_4$ (pH 8.0), 300 mM NaCl, 0.4 mM BME, 1 mM PMSF, 100 µg/mL DNase using a microfluidizer. Lysate was then centrifuged at 16,000 rpm (35,172 × *g*) for 60 min in a Beckman JA-17 rotor chilled to 4°C. Lysate was circulated over 5 mL HiTrap Chelating column (GE Healthcare, Cat# 17-0409-01) charged with 100 mM CoCl$_2$ for 1 hr. Bound protein was then eluted with a linear gradient of imidazole (0–500 mM, 8 CV, 40 mL total, 2 mL/min flow rate). Peak fractions were pooled, combined with TEV protease (75 µg/mL final concentration), and dialyzed against 4 L of buffer containing 50 mM Na$_2$HPO$_4$ (pH 8.0), 300 mM NaCl, and 0.4 mM BME for 16–18 hr at 4°C. Dialysate-containing TEV protease cleaved protein was recirculated for 1 hr over a 5 mL HiTrap Chelating column. Flow-through containing (Gly)$_5$-protein was then concentrated in a 5 kDa MWCO Vivaspin 20 before being loaded on a Superdex 75 (10/300 GL) size-exclusion column equilibrated in 20 mM Tris (pH 8.0), 200 mM NaCl, 10% glycerol, 1 mM TCEP. Peak fractions were pooled and concentrated before snap-freezing in liquid nitrogen.

## DrrA-OCRL

Chimeric 5-phosphatase his6-MBP-(Asn)$_{10}$-TEV-(Gly)$_5$-DrrA(544-647aa)-(Gly)$_5$-OCRL was expressed in BL21 (DE3) *E. coli*. For the proteins described above, bacteria were grown at 37°C in Terrific Broth to an OD$_{600}$ of 0.8. Cultures were shifted to 18°C for 1 hr, induced with 0.1 mM IPTG, and allowed to express protein for 20 hr at 18°C before being harvested. Cells were lysed into 50 mM Na$_2$HPO$_4$ (pH 8.0), 300 mM NaCl, 0.4 mM BME, 1 mM PMSF, 100 µg/mL DNase using a microfluidizer. Lysate

was then centrifuged at 16,000 rpm (35,172 × $g$) for 60 min in a Beckman JA-17 rotor chilled to 4°C. Lysate was circulated over 5 mL HiTrap Chelating column charged with 100 mM $CoCl_2$ for 1 hr. Bound protein was then eluted with a linear gradient of imidazole (0–500 mM, 8 CV, 40 mL total, 2 mL/min flow rate). Peak fractions were pooled, combined with TEV protease (75 µg/mL final concentration), and dialyzed against 4 L of buffer containing 50 mM $Na_2HPO_4$ (pH 8.0), 300 mM NaCl, and 0.4 mM BME for 16–18 hr at 4°C. Dialysate-containing TEV protease cleaved protein was recirculated for 1 hr over a 5 mL HiTrap Chelating column. Flow-through containing (Gly)$_5$-DrrA(544-647aa)-(Gly)$_5$-OCRL were then buffer exchanged into 20 mM HEPES pH 7, 100 mM NaCl, 1 mM DTT using a HiPrep 26/10 desalting column (GE Healthcare, Cat# 17-5087-01). DrrA-OCRL was then loaded onto a 1 mL MonoS (5/50 GL) cation-exchange column (GE Healthcare, Cat# 17-5168-01) equilibrated in 20 mM HEPES pH 7, 100 mM NaCl, 1 mM DTT. DrrA-OCRL was separated from impurities by applying a linear salt gradient (0.1–1 M NaCl) over 45 CV (45 mL total). DrrA-OCRL eluted from the MonoS column in the presence of 250–300 mM NaCl. Peak fractions were pooled and concentrated in a 10 kDa MWCO Vivaspin 6 before being loaded on a Superdex 75 (10/300 GL) size-exclusion column equilibrated in 20 mM Tris (pH 8.0), 200 mM NaCl, 10% glycerol, 1 mM TCEP. Peak fractions were pooled and concentrated before snap-freezing in liquid nitrogen.

## Sortase-mediated peptide ligation

All lipid sensors and catalytic domains were labeled on a N-terminal (Gly)$_5$ motif using Sortase-mediated peptide ligation (*Ton-That et al., 1999*; *Guimaraes et al., 2013*). We devised a novel approach for chemically modifying an LPETGG peptide with fluorescent dyes, which we then conjugated to our protein of interest. The LPETGG peptide was synthesized to >95% purity by ELIM Biopharmaceutical (Hayward, CA) and labeled on the N-terminal amine with N-hydroxysuccinimide (NHS) fluorescent dye derivatives (e.g., NHS-Alexa488). This was achieved by combining 10 mM LPETGG peptide, 15 mM NHS-Alexa488 (or other fluorescent derivatives), and 30 mM triethylamine (Sigma, Cat# 471283) in anhydrous DMSO (Sigma, Cat# 276855). This reaction was incubated overnight in the dark at 23°C before being stored in a –20°C freezer. Prior to labeling (Gly)$_5$-containing proteins, unreacted NHS-Alexa488 remaining in the LPETGG labeling reaction was quenched with 50 mM *tris*(hydroxymethyl) aminomethane (Tris) (pH 8.0) buffer for at least 6 hr. Complete quenching of unreacted NHS-Alexa488 was verified by the inability to label (Gly)$_5$-containing proteins in the absence of a Sortase.

When labeling (Gly)$_5$-containing proteins with the fluorescently labeled LPETGG peptide, we typically combined the following reagents: 50 mM Tris (pH 8.0), 150 mM NaCl, 50 µM (Gly)$_5$-protein, 500 µM Alexa488-LPETGG, and 10–15 µM His$_6$-Sortase (Δ57; lacks first 57 amino acids). This reaction mixture was incubated at 16–18°C for 16–20 hr, before buffer exchange with a G25 Sephadex column (e.g., PD10 or NAP5) to remove majority of dye and dye peptide. The his$_6$-Sortase was then captured on NiNTA agarose resin (QIAGEN) and unbound, labeled protein was separated from remaining fluorescent dye and peptide using a Superdex 75 or Superdex 200 size-exclusion column (24 mL bed volume).

## Preparation of small unilamellar vesicles

The following lipids were used to generate small unilamellar vesicles (SUVs): 1,2-dioleoyl-sn-glycero-3-phosphocholine (18:1 DOPC, Avanti # 850375C), L-α-phosphatidylinositol-4-phosphate (Brain PI(4)P, Avanti Cat# 840045X), L-α-phosphatidylinositol-4,5-bisphosphate (Brain PI(4,5)P$_2$, Avanti # 840046X), 1,2-dipalmitoyl-sn-glycero-3-phosphoethanolamine-N-(lissamine rhodamine B sulfonyl) (16:0 Liss Rhod PE, Avanti # 810158C), and 1,2-dioleoyl-*sn*-glycero-3-phospho-L-serine (18:1 DOPS, Avanti # 840035C). In the main text, 16:0 Liss Rhod PE is referred to as Rhod PE. Lipids were purchased as single-use ampules containing between 0.1 and 5 mg of lipids dissolved in chloroform. Brain PI(4)P and PI(4,5)P$_2$ were purchased as 0.25 mg/mL stocks dissolved in chloroform:methanol:water (20:9:1). To make liposomes, 2 µmole total lipids were combined in a 35 mL glass round-bottom flask containing 2 mL of chloroform. Lipids were dried to a thin film using rotary evaporation with the glass round-bottom flask submerged in a 42°C water bath. After evaporating all the chloroform, the round-bottom flask was flushed with nitrogen gas for at least 30 min. Lipid film was resuspended in 2 mL of PBS (pH 7.2), making a final concentration of 1 mM total lipids. All lipid mixtures expressed as percentages (e.g., 98% DOPC, 2% PI(4)P) are equivalent to molar fractions. For example, a 1 mM lipid mixture containing 98% DOPC and 2% PI(4)P is equivalent to 0.98 mM DOPC and 0.02 mM PI(4)

P. To generate 30–50 nm SUVs, 1 mM total lipid mixtures were extruded through a 0.03 µm pore size 19 mm polycarbonate membrane (Avanti #610002) with filter supports (Avanti #610014) on both sides of the PC membrane. Hydrated lipids at a concentration of 1 mM were extruded through the PC membrane 11 times.

## Preparation of supported lipid bilayers

SLBs are formed on 25 × 75 mm coverglass (IBIDI, #10812). Coverglass is first cleaned with 2% Hellmanex III (Fisher, Cat# 14-385-864) and heated to 60–70°C in a glass coplin jar. It is then incubated for at least 30 min. Coverglass is then washed extensively with Milli-Q water and then etched with Piranha solution (1:3, hydrogen peroxide:sulfuric acid) for 10–15 min the same day and SLBs were formed. Etched coverglass, in water, is rapidly dried with nitrogen gas before adhering to a 6-well sticky-side chamber (IBIDI, Cat# 80608). SLBs are formed by flowing 30 nm SUVs diluted in PBS (pH 7.2) to a total lipid concentration of 0.25 mM. After 30 min, IBIDI chambers are washed with 5 mL of PBS (pH 7.2) to remove non-absorbed SUVs. Membrane defects are blocked for 15 min with a 1 mg/mL beta-casein (Thermo Fisher Scientific, Cat# 37528) diluted in 1× PBS (pH 7.4). Before use as a blocking protein, frozen 10 mg/mL beta-casein stocks were thawed, centrifuged for 30 min at $21,370 \times g$, and 0.22 µm syringe filtered. After blocking SLBs with beta-casein, membranes were washed again with 1 mL of PBS, followed by 1 mL of kinase buffer before TIRF-M.

## Kinetics measurements of PIP lipid phosphorylation

The kinetics of PI(4)P phosphorylation was measured on SLBs formed in IBIDI chambers and visualized using TIRF microscopy. Reaction buffer contained 20 mM HEPES (pH 7.0), 150 mM NaCl, 1 mM ATP, 5 mM $MgCl_2$, 0.5 mM EGTA, 20 mM glucose, 200 µg/mL beta-casein (Thermo Scientific, Cat# 37528), 20 mM BME, 320 µg/mL glucose oxidase (Serva, #22780.01 *Aspergillus niger*), 50 µg/mL catalase (Sigma, #C40-100MG Bovine Liver), and 2 mM Trolox (UV treated, see methods below). Perishable reagents (i.e., glucose oxidase, catalase, and Trolox) were added 5–10 min before image acquisition. For all experiments, we monitored the change in PI(4)P or PI(4,5)$P_2$ membrane density using a solution concentration of 20 nM Alexa647-DrrA(544-647) or 20 nM Alexa488-PLCδ, respectively. We calculated the density of PIP lipids (lipids/µm$^2$) assuming a footprint of 0.72 nm$^2$ for DOPC lipids (*Galush et al., 2008*; *Vacklin et al., 2005*).

## Single-molecule cell lysate assay

Genes encoding mPIP5KA (5KA), hPIP5KB (5KA), hPIP5KC (5KC), zPIP5KA (z5KA), yMss4, and hPIP4K2B (4KB) were cloned into lentiviral expression vectors containing an SFFV promoter to drive expression of N-terminal mNG fusion proteins in mammalian cells. HEK293T cells were transfected with 15 µg of plasmid DNA encoding mNG-tagged lipid kinases, plus 30 µg polyethylenimine (PEI) diluted into 0.5 mL Opti-MEM. Prior to transfection, HEK293T cells were grown to a confluency of 50–60% in 10 cm dishes in DMEM GlutaMAX media (Thermo Fisher, Cat #10566016) containing 10% FBS and penicillin/streptomycin. After 20–24 hr for transfection, adherent HEK293T grown in 10 cm dishes were washed with 5 mL 1× PBS (pH 7.4). After vacuum aspiration of the PBS, cells were incubated in 1 mL of CellStripper (Corning, Cat# 25-056Cl) for 10 min at room temperature. Detached cells were resuspended in 9 mL of PBS and transferred to a 15 mL conical tube. Cells were centrifuged for 5 min at 500 rcf at 4°C. The cell pellet was resuspended in 1 mL of PBS, transferred to a microcentrifuge tube, and centrifuged for 3 min at 500 rcf. After removing PBS by vacuum aspiration, cell pellets were resuspended in 0.6 mL of lysis buffer containing 20 mM HEPES pH 7, 150 mM NaCl, 5 mM $MgCl_2$, Sigma protease inhibitor (Cat# P3840, 1% vol/vol final), Sigma phosphatase inhibitor 2 (0.5% vol/vol final), Sigma PPtase inhibitor 3 (0.5% vol/vol final), 50 mM NaF, 15 µg/mL benzamidine, and 1 mM PMSF. Microtip sonication was used to rupture transfected cell on ice using the following program: 20% amplitude, 2 s ON and 20 s OFF for 15 cycles. Cell lysate was centrifuged for 45 min at 21,300 rcf at 4°C. Following centrifugation, 75% of the supernatant was transferred to new 1.7 mL microcentrifuge tube. The clarified lysate was then mixed 4:1 with lysis buffer containing 50% glycerol (vol/vol). This resulted in lysate containing a final glycerol concentration of 10% (vol/vol). Cell lysate was aliquoted and flash-frozen in liquid nitrogen. We did not observe any difference in the quality of the mNG-labeled protein comparing fresh versus freeze-thawed cell lysate.

To determine the concentration of mNG-tagged lipid kinase in HEK293T cell lysate we generated a twofold serial dilution of bacterially purified mNG diluted in 1× PBS (pH 7.4) and 0.1% NP-40 detergent. The fluorescence intensity was measured using a BioTek 96-well format using a plate reader to generate a standard curve for fluorescence intensity as a function of mNG concentration (*Figure 1— figure supplement 2*). mNG was excited with a 500 nm light using a 500/10 nm bandpass filter. The emission was monitored at 517 nm with high photomultiplier tube (PMT) sensitivity.

## Quantitative fluorescence microscopy using supported lipid bilayer standards

We measured the membrane surface density of Ax647-PIP5KB (*Figure 2*) using previously described methods (*Galush et al., 2008*). In brief, we titrated the molar fraction of Atto655-1,2-dipalmitoyl-sn-glycero-3-phosphoethanolamine (Atto655-DPPE; Atto-TEC, Cat# AD 655-151) against DOPC lipids. SUVs were formed using microtip sonication. We used TIRF microscopy to measure the fluorescence intensity of SLBs containing varying concentrations of Atto655-DPPE. These values were used to generate a standard curve that was used to calculate the surface density of membrane-bound Alexa647-PIP5KB. In order to compare the fluorescence intensity of Alexa647-PIP5KB to Atto655-DPPE, we calculated the scaling factor that accounts for the difference in molecular brightness of the two fluorophores measured on the same microscope with identical camera settings, laser power, filters, and optics. To determine the scaling factor, we measured the fluorescence intensity of solutions containing identical concentrations of either Alexa647-PIP5KB or SUVs containing the same molar concentration of Atto655-DPPE (*Figure 2—figure supplement 3*). These solutions were added to an imaging chamber passivation with a supported membrane (95% DOPC and 5% DOPS) in order to prevent nonspecific absorption of Alexa647-PIP5KB and SUVs containing Atto655-DPPE. To measure the solution intensity of the Alexa647-PIP5KB and Atto655-DPPE-containing samples, we choose a z-axis imaging plane that was 5 μm above the glass surface and acquired at least 20 fluorescence measurements for each solution concentration. These fluorescence intensity values were averaged and plotted as a function of the fluorophore solution concentration. The scaling factor was calculated by dividing the slope of Alexa647-PIP5KB plot by the slope of the Atto655-DPPE plot.

## Surface density calibration of mNG-PIP5KB

Densities of mNG-PIP5KB were estimated using a surface density calibration curve of mNG attached to the lipid bilayer. SLB containing 96% DOPC and 4% Ni-NTA-DOGS by molar percent was incubated with His6-mNG with concentrations ranging from 0.1 nM to 10 nM in 20 mM HEPES (pH 7.0), 150 mM NaCl, 200 μg/mL beta-casein, 5 mM BME for 30 min. The chambers were rinsed with 1 mL of buffer and a calibration curve was established between the TIRF average intensity and surface densities measured by FCS of mNG on the membrane using previously described methods (*Chung et al., 2019*; *Chung et al., 2018*). The same calibration curve was used to estimate the surface density of mNG-PIP5KB. The kinetic measurements were performed on SLB containing 96% DOPC and 4% PI(4)P by molar percent, with the presence of 20 nM Cy3-PLCδ to monitor the change of PI(4,5)P$_2$. The total PIP lipid was calculated to be at 55,555 lipids/μm$^2$. The start and end of the reaction were approximated to have 0 and 55,555 lipids/μm$^2$ of PI(4,5)P$_2$, respectively. Images were acquired at a 2 s interval. PI(4,5)P$_2$ and mNG-PIP5KB surface density was calculated for each time point. The $\frac{d\mathrm{PI}(4,5)\mathrm{P2}}{dt}$ for each time point was obtained by using the slope of linear regression of PI(4,5)P$_2$ level change in a +2 s to –2 s time range.

## Feedback analysis of PIP5K

Alexa488-PLCδ intensity was measured from TIRF images acquired at a 2 s interval. The reaction coordinate (*x*) for each time point was calculated by normalizing the start intensity to 0 and end intensity to 1. The $\frac{dx}{dt}$ for each time point was obtained by using the slope of linear regression of reaction coordinate change in a +2 s to –2 s time range.

## Chromium-patterned glass coverslips

25 × 75 mm No. 1.5 thickness glass coverslips were cleaned in acetone by sonication, then washed with Milli-Q water extensively. The coverslips were dried by nitrogen gas, then baked on 120°C hot plate for 5 min. S1805-positive photoresist were spin-coated on the coverslips by spinning for 2 s at

500 rpm (ACL 440), then for 30 s at 4111 rpm (ACL 3900). The photoresist on the edge of the coverslips were removed by cotton swap soaked with acetone, then baked on 120°C hot plate for 1 min. Mask with desired pattern was mounted on an OAI Series 200 Aligner. The photoresist-coated coverslip was exposed for 0.6 s with UV power around 30 mJ, then developed with MicroPosit MF-321 Liquid Developer for 40 s with mild shaking. The developed coverslips were rinsed with water and dried with nitrogen gas. Approximately 9-nM-thick chromium was subsequently deposited on the coverslips using an electron beam evaporator at $1 \times 10^{-6}$ torr. Finally, photoresist is lifted from chromium-patterned glass substrates by bath sonication in Dow Electronic Materials MicroPosit Remover 1165 for 10 min two times, then washed with water.

## Microscope hardware and imaging acquisition

Single-molecule imaging experiments were performed on an inverted Nikon Eclipse Ti and Ti2 microscopes using a ×100 Nikon objective (1.49 NA) oil immersion TIRF objective. The macroscopic spatial patterning of PIP compositional patterns and the chromium-patterned SLB were visualized using a ×60 Apo TIRF oil immersion objective (1.45 NA). The x-axis and y-axis positions were manually controlled using an ASI stage and joystick. All images were acquired using either a iXon Ultra or iXion Life 897 EMCCD camera (Andor Technology Ltd, UK). Fluorescently labeled proteins were excited with either a 488 nm, 561 nm, or 637 nm diode laser (OBIS laser diode, Coherent Inc, Santa Clara, CA) controlled with either a Solemere (Nikon Ti) or Vortran (Nikon Ti2) laser drive with acousto-optic tunable filter (AOTF) control. The power output measured through the objective for single-particle imaging was 1–3 mW. For dual-color imaging of spatial PIP lipid patterns on SLBs, samples were excited with 0.2–0.5 mW 488 nm and 0.2–0.5 mW 637 nm light, as measured through the objective. Excitation light was passed through the following dichroic filter cubes before illuminating the sample: (1) ZT488/647rpc and (2) ZT561rdc (ET575LP) (Semrock). Fluorescence emission was detected on an ANDOR EMCCD camera position after a Sutter emission filter wheel housing the following emission filters: ET525/50M, ET600/50M, ET700/75M (Semrock). All experiments were performed at room temperature (23°C). Microscope hardware was controlled using both Micro-Manager v4.0 (*Edelstein et al., 2010*) and Nikon NIS elements.

## Single-particle tracking

Fluorescent particle detection and tracking was performed using the ImageJ/Fiji TrackMate plugin (*Jaqaman et al., 2008*). Image stacks containing ~1000 16-bit image stacks in the form of a .nd2 file or .tif stack were loaded in ImageJ. Image sequences were cropped to 400 × 400 pixels in order to minimize differences in field illumination caused by TIRF illumination. Using the LoG detector option, particles were identified based on brightness and their signal-to-noise ratio. After identifying the position of all fluorescent particles, we used the LAP tracker to generate particle trajectories that followed molecular displacement as a function of time. Particle trajectories were then filtered based on Track Start (removed trajectories that began in first frame), Track End (removed trajectories present in last frame), Duration (removed trajectories ≤2 frames), Track displacement (removed immobilized particles), and X – Y location (removed particles near the edge of the images). Trajectories were filtered to remove 1–5% of particles that were immobilized throughout the image sequence. The TrackMate output files were analyzed using Prism to calculate the single-molecule dwell times and diffusion coefficients.

Step-size distribution of single-particle trajectories was plotted in Prism. For all analyses presented in this article, the bin size for the step-size distribution equals 0.01 µm. For curving fitting, the step-size distributions were plotted as probability density versus step size (µm). This was achieved by dividing the frequency distribution (i.e., y-axis values) by the bin size (0.01 µm). The probability density versus step-size plots were fit to the following one- or two-species distributions:

Single-species model:

$$f(r) = \frac{r}{2D\tau} e^{-\left(\frac{r^2}{4D\tau}\right)}$$

Two-species model:

$$f(r) = \alpha \frac{r}{2D_1\tau} e^{-\left(\frac{r^2}{4D_1\tau}\right)} + (1 - \alpha) \frac{r}{2D_2\tau} e^{-\left(\frac{r^2}{4D_2\tau}\right)}$$

Variables are defined as the D1 = diffusion coefficient species 1 ($\mu m^2$/s), D2 = diffusion coefficient species 2 ($\mu m^2$/s), alpha ($\tau_1$ = % of species 1, $r$ = step size ($\mu$m)), $\tau_2$ = time interval between steps (s). Final step-size distribution plots were generated in Prism graphing software and using the following equations: (1 species model): f(x) = x/(2*D1*t)*exp(-(x^2/ (4*D1*t))), (2 species model): f(x) = alpha*(x/ (2*D1*t)*exp(-(x^2/ (4*D1*t)))) + (1-alpha)*(x/(2*D2*t)*exp(-(x^2/ (4*D2*t)))).

During our characterization of membrane-bound mNG-PIP5K and Ax647-PIP5K, we established conditions that caused fluorescently labeled PIP5K to diffuse more slowly. This includes the use of expired or oxidize PI(4,5)P$_2$ lipids, as well as membrane blocking with 1 mg/mL beta-casein for >15 min. For both sets of conditions, we measured diffusion coefficients for mNG-PIP5K and Ax647-PIP5K that were 0.09–0.12 $\mu m^2$/s. For the data presented in this article, we controlled for these factors and measured faster diffusion coefficients in the range of 0.14–0.19 $\mu m^2$/s.

To calculate the single-molecule dwell times for Ax647-PIP4K and Ax647-PIP5K, we generated a cumulative distribution frequency (CDF) plot using the frame interval as the bin size (e.g., 50 ms). The $\log_{10}$(1-CDF) was plotted against the dwell time and fit to either a single- or double-exponential decay curve.

Single-exponential model:

$$f(t) = e^{(-x/\tau)}$$

Two-exponential model:

$$f(t) = \alpha * e^{(-x/\tau 1)} + (1 - \alpha) * e^{(-x/\tau 2)}$$

Fitting procedure was initiated with a single exponential. In case of a low-quality single-exponential fit, a maximum of two-species model was used. For double-exponential fit, alpha (α) represents the fraction of fast-dissociating molecules characterized by $\tau_1$.

## Quantification of mNeonGreen fluorescence brightness

Supported membranes containing 4% PI(4,5)P$_2$ and 96% DOPC lipids were used to recruit mNG-tagged PIP4K and PIP5K proteins from solution to bilayers. Proteins were excited and visualized by TIRF microscopy using ~1 mW of 488 nm laser power measured through a ×100 Nikon objective (1.49 NA) oil immersion TIRF objective. To determine the molecular brightness distribution, images were acquired from N = 10–30 fields of views. Membrane-bound mNG-tagged kinases were not exposed to 488 nm light until acquiring a single snapshot. This ensured that a negligible amount of photo-bleaching occurred prior to imaging. Unless stated otherwise, quantification of molecular brightness corresponds to molecules that had not been continuously imaged.

### Image analysis, curve fitting, and statistics

Image analysis was performed using ImageJ. Curve fitting was performed using Prism 9 (GraphPad). Single-molecule dwell time, step size, and molecular brightness distributions presented in this article represent combined data from three technical replicates with 2–3 movies acquired from multiple fields of view for each experimental condition. Dwell time distributions and curve fits were generated with N ≥ 1000 particle trajectories (*Figure 2B*, *Figure 3C–E*, *Figure 2—figure supplement 2*). Step-size distribution plots and curve fits represent ≥10,000 measured displacements (*Figure 1J*, *Figure 2C*, *Figure 3F*, *Figure 3—figure supplement 1*). Single-molecule brightness distribution plots were generated from analyzing N ≥ 1000 fluorescent particles (*Figure 1F–I*, *Figure 4C, D, G, and H*, *Figure 1—figure supplement 3*, *Figure 1—figure supplement 4D and E*, *Figure 4—figure supplement 1*). See *Tables 1 and 2*, and figure legends for more exact statistics associated with the data presented.

*Supplementary file 1*

## Acknowledgements

We thank members of the Hansen lab for critical reading of our manuscript. Funding was provided by NIH grant P01 AI091580 (JTG), by the Novo Nordisk Foundation Challenge Programme as part of the Center for Geometrically Engineered Cellular Systems (JTG), National Research Service Award

Postdoctoral Fellowship (SDH, F32 GM111010-02), University of Oregon Start-up funds (SDH), NSF CAREER Award (SDH, MCB-2048060), and Molecular Biology and Biophysics Training Program (BRD, NIH T32 GM007759).

## Additional information

### Funding

| Funder | Grant reference number | Author |
| --- | --- | --- |
| National Science Foundation | CAREER MCB-2048060 | Scott D Hansen |
| University of Oregon, Department of Chemistry and Biochemistry | lab startup funds | Scott D Hansen |
| Novo Nordisk Foundation Challenge Programme | Center for Geometrically Engineered Cellular Systems | Jay T Groves |
| National Institute of Health, NIGMS | National Research Service Award (NRSA) | Scott D Hansen |
| National Institute of Health, NIGMS | T32 GM007759 | Benjamin R Duewell |
| National Institute of Health, NIGMS | F32 GM111010-02 | Scott D Hansen |

The funders had no role in study design, data collection and interpretation, or the decision to submit the work for publication.

### Author contributions

Scott D Hansen, Conceptualization, Resources, Data curation, Formal analysis, Supervision, Funding acquisition, Validation, Investigation, Visualization, Methodology, Writing – original draft, Project administration, Writing – review and editing; Albert A Lee, Data curation, Formal analysis, Validation, Investigation, Visualization, Methodology, Writing – review and editing; Benjamin R Duewell, Resources, Investigation, Methodology, Writing – review and editing; Jay T Groves, Conceptualization, Supervision, Funding acquisition, Writing – review and editing

### Author ORCIDs

Scott D Hansen http://orcid.org/0000-0001-7005-6200
Jay T Groves http://orcid.org/0000-0002-3037-5220

### Decision letter and Author response

Decision letter https://doi.org/10.7554/eLife.73747.sa1
Author response https://doi.org/10.7554/eLife.73747.sa2

## Additional files

### Supplementary files

• Transparent reporting form

• Supplementary file 1. Plasmids and recombinant proteins. Description of plasmids used for recombinant protein expression, purification, and transient transfection.

### Data availability

All data generated or analysed during this study are included in the manuscript and supporting file; Source Data files have been provided for all figures.

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
