## [Editor Report]

This study presents highly interesting and detailed biochemical analyses into the mutual relationship between PI(4,5)P_2_ lipids and their kinase PIP5K, which engage in an exciting pattern-forming reaction on membranes. Using direct single-molecule imaging approaches, the authors find cooperative recruitment of PIP5K to the membrane, dimerization-enhanced catalytic efficiency and indications of allosteric regulation. This article is of interest to a wide range of readers who study the biology of lipid-modifying enzymes, especially as it relates to interfacial reaction kinetics in biological membranes.

---

## [Decision Letter]

**Decision letter after peer review:**

Thank you for submitting your article "Membrane-mediated dimerization potentiates PIP5K lipid kinase activity" for consideration by *eLife*. Your article has been reviewed by 3 peer reviewers, including Janice L Robertson as the Reviewing Editor and Reviewer #1, and the evaluation has been overseen by Philip Cole as the Senior Editor.

Essential revisions:

In general, the reviewers found this paper to be of high interest, presenting exciting results leading to a mechanistic understanding of regulation of pattern forming behavior by PIP5K kinase in membranes. While the data presented was mostly convincing, there are concerns about the interpretation of whether this is dimer specific or related to higher-order oligomerization. In addition, the study suffers from a lack of clarity in the writing, as well as inconsistencies and shortcomings in data presentation. Specifically, the following revisions are required:

1) While there is evidence of linkage to oligomerization, the overall interpretation that the mechanism is specifically linked dimerization is not strongly supported. To address this, the studies that are currently presented require additional clarification, and control experiments are necessary to map out the dimer specific signal on membranes. Alternatively, the mechanism may be modified to consider that this effect may come from an ensemble of higher-order oligomeric states. With this, the paper and the title require significant revisions to clarify this point.

2) Sample sizes, errors and statistical tests must be calculated and clearly presented in the paper and in the figure legends. In its current form, it is not possible to assess whether the results presented are reproducible.

3) The writing needs significant revisions to improve coherency and clarity. The manuscript should be revised to improve accessibility for a broad group of readers, with improved background and discussion of the physical concepts and equations that are presented. In addition, the discussion should be revised to better related the current findings to recently published studies.

4) Address the specific comments and questions raised by the reviewers below.

Overall, the study would benefit from editing to improve coherency and clarity.

*Reviewer #1 (Recommendations for the authors):*

1. The results presented in the size exclusion chromatography analysis (Figure 1B) need some clarification. There is a hypothesized dimer peak for PIP4K, monomer peak for zPIP5KA/D84R and a sharp monodispersed peak for zPIP5KA, that is interpreted as a monomer/dimer mix. While rapid monomer-dimer exchange might account for this, it is surprising that such as mixture is not smeared over both monomer and dimer elution volumes. Changes in column behavior can also create the appearance of peak shifts and so additional calibration is necessary in order to be able to interpret this data. The protein samples should be run with a calibration sample that can control for changes in column behavior and also allow the identification of expected peak positions. In addition, the protein concentration prior to loading should be controlled, as this can also lead to observation of different oligomeric species. An alternate method that often presents clearer results is cross-linking the dilute protein sample by something like glutaraldehyde and the running on SDS-PAGE. This may be used to rule out higher-order oligomeric species.

2. The brightness analysis presented in Figure 1D,E indicates either that the mNG is folded and fluorescent with 100% yield, or that the PIP4KB being observed is a higher order oligomer than dimer. The PIP4KB intensity distribution in Figure 1E shows very little overlap with the monomeric distributions, and a broad intensity distribution that is 2-3 fold greater that the monomeric samples. Considering binomial labeling probabilities, with around 80% labeling efficiency, a dimer is expected to yield nearly equal probabilities of single and double labelled species, which does appear to be the case for the fluorophore labelled protein presented in Figure 1 —figure supplement 2. For the mNG samples, it is rare for the fluorophore to fold with 100% efficiency, and so it is more probable that the resultant intensity distribution corresponds to > 2 proteins. Photobleaching analysis of the spots would reveal clearer information and test whether higher-order oligomeric species are present.

3. The statement of weak monomer-dimer equilibrium is not strongly supported by the presented data. To say this, one would need to show a reversible monomer to dimer conversion as a function of protein density, such as an association isotherm. While the data presented shows clear evidence of monomers, the change in behavior with protein concentration is in line with higher order oligomerization. This is even diagrammed in Figure 2A. To isolate the dimer-specific species signal, functional cross-linking of the protein might be useful to switch the protein from monomeric to dimeric states and characterize the dimer-specific signal at low protein densities.

4. The major argument for the effects being dimerization dependent are from the results of the D51R mutation, as it is interpreted that this mutation solely affects dimerization. However, it seems plausible that other things may change in the protein that impact catalytic activity that are outside of the dimerization. It is unclear as to whether this particular mutant is well folded on its own, thus further analysis of the protein would be useful here. Is the binding and catalytic activity preserved for D51R in solution, and how does this compare to wild-type under monomeric conditions? If dimerization was observable, for example via AUC, then this would also provide supportive evidence of the fold, as well as report the change in stability.

5. The binding curves for PIP5KA to the membrane in Figure 2F are very clear and present interesting results on the different binding capacities as a function of PIP2. Since the lower leg of the monomer-oligomer equilibrium reaction on the membrane (Figure 4D) will be dependent on the membrane associated protein density, it is important to have this full binding relationship for all constructs in order to interpret the results from the different experiments. As such, this information seems essential for the interpretation of the D51R data. While the dwell time and diffusion distributions show similar behavior to WT, this only reports on the behavior of the molecules that are bound and are all likely to be monomers under the examined conditions (Figure 3C,D). It does not directly report on whether there is the same amount of D51R vs. WT on the membrane. In this case, the data presented in Figure 5, is not expected to have the same amount of membrane bound PIP5K, which could account for the differences in behavior rather than dimerization. Indeed, the addition of 1 μm of D51R increases the observation of the compositional patterns.

*Reviewer #2 (Recommendations for the authors):*

1. It is not a weakness but rather a limitation that the study is not extended experimentally into real cell situations and only speculates about the potential impact of its finding to the control of the enzymes in real cells.

2. Some of the methods and terms should be defined and explained for the general reader: e.g. It would be useful to define "dwell time" or explain what the Y axis is in Figure 2B. (for example what is CDF?).

3. Similarly, the detailed description of how the catalytic activity of the enzymes were calculated (lines 270-310) is important for the experts. However, it would be desirable to "translate" the following sentences to the general reader.

"When the derivative of the PIP5K kinase reaction traces were plotted against the reaction coordinate, x, the curve displayed a high degree of asymmetry that required a second-order term to fit (kWT(x) = k0 + k1x + k2x2) (Figure 4E). In contrast, the dx⁄dt curve for the dimer mutant was parabolic and could be fit using an equation that describes an enzyme with first order positive feedback (kD51R(x) = k0 + k1x) based on product binding (Figure 4E)." This may not be necessary if the paper is published in a biophysical journal, but for most of the readers of *eLife*, this description is not very accessible.

4. The legend to Figure 4 Supplement 1. does not really describe what is measured here. First of all, what is measured as "auto-correlation" should be explained. It is not clear what the different colors mean (the concentrations refer to what component?). Panel A-C appear to show graphs on the various PIP5KB forms (wt or mutants), but the Figure legend for panel A does not refer to PIP5KB.

5. The same is true for the legend to Figure 4 Supplement 2. From the description, one would expect to see three similar measurements for the three forms of the enzyme. Clearly, panel A shows a curve that is not explained. Panels B and C show FCS density as a function of TIRF intensity. What is measured here (what is FCS density?)? How do these curves inform about lipid kinase activity? Clearly, the Authors should do better to explain what is shown here and how does that translate to kinase activity.

6. I think the DrrA designation of what is better known as SidM should at least be clearly stated somewhere (preferably also in the Figure legends).

7. Remove the prime from all "5'-phosphatase". There are no prime numbers on the inositol ring just simple numbers designating the positions on the single carbon ring.

*Reviewer #3 (Recommendations for the authors):*

1. Line 150 and Figure 1B: The authors are referring to a dimer mutant of PIP5K. They need to clarify if that means that the protein is locked in its dimerized state or if the dimerization interface is disrupted and thus the protein resides in its monomeric state. If, as indicated by the 'dimer mutant' reference the protein should be locked in a dimer state, why does the SEC elution profile and their annotation rather indicate that the protein elutes as a monomer? Also, it would be beneficial for the reader to indicate the name of the dimer mutant in the text.

2. Figure 1 Supplement 2: Monomer/dimer distribution of mNG-PIP4K in Figure 1E are considerably different to the ones shown for the Alexa488-PIP4K version in supplement 2. Is there a reason for this comparison, especially as the authors are later introducing another chemical label with the Alexa647-conjugation construct? They should rather repeat the experiment with the Alexa647-conjugation construct that they are using later on.

3. Figure 1B: In Figure 1B, the 'dimer' mutant of PIP5K is annotated as zPIP5KA(D84R), however in the respective figure caption, it is referred to as zPIP5KA(D51R) mutant – which one now is the correct mutant? The authors should use a size standard to calibrate the SEC elution profiles to ensure the correct detection of monomers and dimers (SEC-MALS would be even better suited). The critical reader would furthermore like to see an SDS-PAGE gel of the purified proteins, as especially the elution profile of PIP5K indicates impurities.

4. Figure 1C/D: The authors should indicate the abbreviations of both proteins in the respective figure captions.

5. How do the authors account for bleaching in their TIRF single particle tracking assay?

6. Supplementary Figure 1A: What is the reproducibility of the mNG-calibration curve for the quantitative analysis of kinase presence in the cell lysate? As indicated in the figure caption, only a representative curve is shown, however adding standard errors for each concentration and the indication of the number of replicates would need to be included.

7. Lines 180-181: Several mNG-fusion constructs are named, but it is completely unclear for the reader what they are, as they have not been introduced (e.g. mNG-PIP5KA/B/C or mNG-yMss4?). Even more importantly, in Figure 1E, which is the figure that has been referenced in the text, there are again other abbreviations for the used constructs that do not fit the ones named in the text or have not been referenced in the text. Likewise, also no consistency between the abbreviations between Figures 1E and F. the authors need to clarify this and keep the names of constructs consistent throughout the whole text, figures, and supplementary information.

8. Figure 1F: The authors are referring to a step size distribution in the figure caption, but at the same time state diffusion coefficients for the respective constructs – they need to elaborate on how exactly they got the diffusion coefficients from the step size distribution, more specifically: what kind of fitting they used. As in Figure 2C, the fit should also be included in the plot. Furthermore, what is the error and standard deviation for the diffusion coefficients? It would be beneficial for the reader to also state the deviations in lines 190/191.

9. As stated in line 192, the authors are comparing the diffusion coefficients of the cell lysate kinases with purified kinases in Figure 2C. However, in the two cases different membrane compositions have been used – in Figure 1F they state that 2 % PIP2 have been used, while in Figure 2C 4 % have been incorporated. Why do they change the proportion of PIP2 in the membrane between the two cases? This inconsistency especially impedes their assumption in line 193, that the membrane binding of kinases in the cell lysate is indistinguishable from the purified case.

10. Figure 2B: For readers that are not familiar with single particle tracking, the cumulative density function should be introduced in the caption of the figure or even in the axis label. Same comment for Figure 3CDE.

11. Figure 2C: Again, a new abbreviation is being introduced for PIP5KB.

12. Figure 2F: Error bars are missing and SEM of Kd values. In the respective caption, they should state the number of independent replicates. They also need to indicate in the caption that the nh – values they are referring to in the figure are the Hill coefficients. The manuscript would benefit considerably if they would repeat the experiments shown in this Figure also with the D51R mutant – especially as they are comparing the mutant with the wild type throughout the manuscript, and only for Kd values the respective differences cannot be judged. Similar comment for Figure 4AB regarding error bars and replicate numbers.

13. Figure 3CD: The authors need to indicate in the caption or the figure that this is a low protein density condition.

14. Figure 3F: Please state the respective diffusion coefficients you have obtained from the used fit. Same comment for Supplementary Figure 3 (and adjust x-axis label to be consistent with similar graphs).

15. Supplementary Figure 4 (Figure supplement 1 and 2): It seems like the Figure captions of Supplementary Figure 4 Supplement 1 and 2 have been mistaken for each other – please readjust.

16. Supplementary Figure 4 (Figure supplement 1): In panel B apparently different protein concentrations have been used compared to panels A and C – why?

17. Supplementary Figure 4 (Figure supplement 2): Panels BC – Please indicate error bars and replicate numbers.

18. Figure 4A: Why don't the authors compare the reaction half-times of wild-type and the D51R mutant at similar concentrations to enable a more direct comparison between the two? Are the reaction half-times based on the kinetic traces seen in the respective supplementary Figure? If yes, why is there such a difference in the reaction half-time between the trace at 1 nM PIP5K in the supplement (roughly 160 seconds) and the 280 seconds stated in the main Figure? Why are the traces for higher PIP5K concentrations not included in the supplement?

19. Figure 2 and Figure 4: Why is the fluorescence of a membrane dye used to calibrate for PIP5K membrane density in Figure 2, whereas a more sensitive FCS-approach is used in Figure 4 to do the same? They should have likewise measured FCS for the Alexa-fusion construct in Figure 2.

20. Lines 294-301: in Line 296, the authors give a definition of the reaction coordinate x. However, they are not defining the omega factor that they are using, and it is not clear what they mean with PIP1 and PIP2 – should that be the educt and product? Please clarify and annotate in the text.

21. Lines 324-325: Please add the information that the DrrA-OCRL construct is the opposing 5`-phosphatase.

22. Figure 5B: What is meant with 0`-9`? Are these frame numbers (if yes what is the time difference between them? If the numbering should indicate time – what unit is it? Same question for Figure 6A).

[Editors' note: further revisions were suggested prior to acceptance, as described below.]

Thank you for resubmitting your work entitled "Membrane-mediated dimerization potentiates PIP5K lipid kinase activity" for further consideration by *eLife*. Your revised article has been evaluated by Jonathan Cooper (Senior Editor) and a Reviewing Editor.

The manuscript has been improved but there is one remaining issue that needs to be addressed, as outlined below:

Regarding the text starting on line 375:

"Conversely, the phosphorylation rate constant for wild-type PIP5KB began with a slow rate and then transitioned to a rate of 2.7x10-3 lipids/μm2•sec per kinase as the reaction progressed."

However, when examining the plots in Figure 5G, it is not clear that there is a slow component to the rate. If the slower rate is in the region of the graph close to the origin, then this area should be highlighted and zoomed in with an inset plot showing an expanded scale. Otherwise, this statement is not corroborated with the data as shown.

---

## [Author Response]

Essential revisions:In general, the reviewers found this paper to be of high interest, presenting exciting results leading to a mechanistic understanding of regulation of pattern forming behavior by PIP5K kinase in membranes. While the data presented was mostly convincing, there are concerns about the interpretation of whether this is dimer specific or related to higher-order oligomerization. In addition, the study suffers from a lack of clarity in the writing, as well as inconsistencies and shortcomings in data presentation. Specifically, the following revisions are required:1) While there is evidence of linkage to oligomerization, the overall interpretation that the mechanism is specifically linked dimerization is not strongly supported. To address this, the studies that are currently presented require additional clarification, and control experiments are necessary to map out the dimer specific signal on membranes. Alternatively, the mechanism may be modified to consider that this effect may come from an ensemble of higher-order oligomeric states. With this, the paper and the title require significant revisions to clarify this point.

Our revised manuscript contains a significant amount of new data demonstrating that fluorescently labeled PIP5K and PIP4K can bind to membranes as either monomers or dimers, respectively, at low surface densities. This is supported by control experiments analyzing the stepwise photobleaching of single membrane bound particles of mNG-PIP5K and mNG-PIP4K. Consistent with SEC data, our single molecule experiments do not show any evidence of high-order oligomers forming on supported membranes. In the Discussion section, we speculate that high-order oligomers can potentially form. However, there is currently no direct evidence to support the formation of high-ordered PIP5K oligomers. If such a mechanism exists, future structural biochemistry studies will be necessary to identify a new protein-protein binding interface that mediate this unknown interaction.

During our revision, we established new experimental conditions to directly visualize membrane mediated dimerization of mNG-PIP5KB monomers, while still being able to spatially resolve single molecules (see Figure 4). The small fraction of dimeric mNG-PIP5KB species observed on membranes is absent from the population when we characterize the dimerization deficient mutant, PIP5KB (D51R). The efforts to directly visualize mNG-PIP5KB were made possible by a graduate student, Benjamin Duewell. In our revised manuscript, Benjamin is now included as a contributing author.

The reviewers suggested that we modify the title of our manuscript to state that PIP5K activity is potentiated by higher order oligomerization of PIP5K, rather than dimerization. Based on our additional control data, we more convincingly demonstrate that membrane-mediated dimerization is the mechanism that enhances PIP5K lipid kinase activity. Hopefully our revisions clarify the issues raised in our original submission and that the reviewers agree with our overall conclusions.

2) Sample sizes, errors and statistical tests must be calculated and clearly presented in the paper and in the figure legends. In its current form, it is not possible to assess whether the results presented are reproducible.

Details about sample size and statistics were located in the Methods section of our original submission. In our revised manuscript, a more comprehensive description of sample size and statistics is included in the figure legends, Table 1, Table 2, and the main text.

3) The writing needs significant revisions to improve coherency and clarity. The manuscript should be revised to improve accessibility for a broad group of readers, with improved background and discussion of the physical concepts and equations that are presented. In addition, the discussion should be revised to better related the current findings to recently published studies.

We appreciate the reviewer comments that our writing needed improvement to maximize accessibility and readership. In our revision, we expanded descriptions of our methodology and rationale for different experiments. We describe the limitation of some approaches (e.g. FCS) and explain why an alternative approach was necessary. The addition of control data should also improve the clarity of our results and conclusions.

4) Address the specific comments and questions raised by the reviewers below.

We addressed all the review comments, point by point. In our rebuttal, we have also copied and pasted textual revisions below specific reviewer comments. For readability, minor typos and adjustments to figure numbering were corrected, but not highlighted in the main text.

Overall, the study would benefit from editing to improve coherency and clarity.Reviewer #1 (Recommendations for the authors):1. The results presented in the size exclusion chromatography analysis (Figure 1B) need some clarification. There is a hypothesized dimer peak for PIP4K, monomer peak for zPIP5KA/D84R and a sharp monodispersed peak for zPIP5KA, that is interpreted as a monomer/dimer mix. While rapid monomer-dimer exchange might account for this, it is surprising that such as mixture is not smeared over both monomer and dimer elution volumes. Changes in column behavior can also create the appearance of peak shifts and so additional calibration is necessary in order to be able to interpret this data. The protein samples should be run with a calibration sample that can control for changes in column behavior and also allow the identification of expected peak positions. In addition, the protein concentration prior to loading should be controlled, as this can also lead to observation of different oligomeric species. An alternate method that often presents clearer results is cross-linking the dilute protein sample by something like glutaraldehyde and the running on SDS-PAGE. This may be used to rule out higher-order oligomeric species.

The reviewer makes a good point concerning the limitations of our SEC analysis. We also agree that it’s surprising the zPIP5KA monomer/dimer mix is not distributed over a broad peak. The main purpose of the SEC data included in our manuscript was to confirm previously reported results from Hu et al. (2015), who reported a similar elution profile to ours. Since our SEC data is confirmation of published results and less conclusive compared to our single molecule imaging experiments, we moved the SEC data from Figure 1 to the Figure 1 —figure supplement 1 in our revised manuscript. To strengthen our SEC data, we ran several protein molecular weight standards over our Superdex 200 column, which we compare to the elution profiles of PIP4KB, zPIP5KA, and zPIP5KA(D84R). In addition, we purified more mNG-PIP5KB (wild type and D51R) and compared their elution profiles on a Superdex 200 column (Figure 4A). To rule out the presence of higher order oligomers we performed single molecule photobleaching experiments (Figure 1).

Overall, we conclude that wild type and dimerization deficient PIP5K mutants exhibit distinct SEC elution profiles. We and other groups have interpreted this result as representing either the monomeric or dimeric states of PIP5K. However, when mNG-PIP5K is diluted to a sub-nanomolar concentration and characterized by smTIRF microscopy, it appears to be exclusively monomeric based on our molecular brightness analysis. By contrast, when PIP4K is diluted to a sub-nanomolar concentration, it binds to supported membranes exclusively as a dimer. This supports a model in which PIP5K exists in a monomer-dimer equilibrium, while PIP4K is a constitutive dimer. Unfortunately, we and others have not been able to measure the K_D_ for PIP5K dimerization in solution. This challenge potentially reflects the weak affinity between PIP5K monomers.

2. The brightness analysis presented in Figure 1D,E indicates either that the mNG is folded and fluorescent with 100% yield, or that the PIP4KB being observed is a higher order oligomer than dimer. The PIP4KB intensity distribution in Figure 1E shows very little overlap with the monomeric distributions, and a broad intensity distribution that is 2-3 fold greater that the monomeric samples. Considering binomial labeling probabilities, with around 80% labeling efficiency, a dimer is expected to yield nearly equal probabilities of single and double labelled species, which does appear to be the case for the fluorophore labelled protein presented in Figure 1 —figure supplement 2. For the mNG samples, it is rare for the fluorophore to fold with 100% efficiency, and so it is more probable that the resultant intensity distribution corresponds to > 2 proteins. Photobleaching analysis of the spots would reveal clearer information and test whether higher-order oligomeric species are present.

This is a good point raised by the reviewer. We believe that part of the confusion concerning our interpretation of PIP4K/PIP5K oligomerization stemmed from the way in which data was *originally plotted* in Figure 1E (i.e. 3D plot with a tilted x-axis). We now provide more extensive experimentation showing that we only observe membrane bound monomers and dimers in our single molecule imaging experiments. Based on our additional characterization, there is no evidence supporting the formation of higher order PIP4K or PIP5K oligomers.

Based on our spectrophotometry measurements, mNG and mNG-PIP5KB purified from either bacteria or insect cells contain ~80% mature and fluorescent chromophores. This is calculated based on the fraction of purified mNG protein that absorbs light at 517 nm, compared to the total protein concentration calculated based on absorption at 280 nm. Knowing this, a protein that is constitutively dimeric (i.e. PIP4K) will exist as a population of molecules with either 1 or 2 mature mNG proteins when visualized using single molecule TIRF-M. Supporting this notion, we performed single molecule step-photobleaching experiments of mNG-PIP4KB (Figure 1D). By monitoring the maximum pixel intensity of single mNG-PIP4KB molecules, we observed that ~80% of the protein contain 2 mature mNG. We never observed mNG-PIP4KB particles photobleached in more than 2 steps (Figure 1D). By contrast, mNG-zPIP5KA, mNG-PIP5KA, and mNG-PIP5KB all photobleached in a single step (Figure 1E and Figure 1 —figure supplement 4). To quantify changes in molecular brightness during step-photobleaching, we continuously imaged single mNG-PIP4K and mNG-PIP5K particles. In Figure 1F-1G, we show how the distribution of mNG-PIP4K molecular brightness shifts from bimodal to unimodal over the course of the 10 second image acquisition. Conversely, the distribution of mNG-PIP5K molecular brightness remains unimodal because particles always photobleached in a single step. Plotted in Figure 1H, we show how the proportion of mNG-PIP4K and mNG-PIP5K with either 1 or 2 visible mNG molecules changes as a function of time from the beginning of image acquisition. Also see Video 1.

3. The statement of weak monomer-dimer equilibrium is not strongly supported by the presented data. To say this, one would need to show a reversible monomer to dimer conversion as a function of protein density, such as an association isotherm. While the data presented shows clear evidence of monomers, the change in behavior with protein concentration is in line with higher order oligomerization. This is even diagrammed in Figure 2A. To isolate the dimer-specific species signal, functional cross-linking of the protein might be useful to switch the protein from monomeric to dimeric states and characterize the dimer-specific signal at low protein densities.

Previously published research established that zPIP5KA is a dimer in solution based on size exclusion chromatography and multiangle light scattering (Hu et al. 2015). However, when we dilute zPIP5KA and other PIP5K family proteins to sub-nanomolar concentrations, mNG labeled kinases associate with PI(4,5)P_2_ containing membranes primarily as monomers. This result implies that PIP5K must be in a weak monomer-dimer equilibrium. By contrast, dilution of mNG-PIP4KB to a sub-nanomolar concentration does not cause the kinase to dissociate into monomers. Based on our improved single molecule brightness analysis (see Figure 1), there no evidence of higher order oligomers according to the smTIRF measurements.

In our revised manuscript, we include a new Figure 4 showing the direct visualization of mNG-PIP5KB dimer formation on supported membranes.

In our revised manuscript, we added a comment in legend Figure 2A to emphasize that the cartoon is not drawn to imply a particular oligomerization state. We’re open to suggestions if the reviews have any ideas for illustrating an increase in protein surface density without implying a specific oligomerization state.

Chemical cross-linking experiments is a great idea. However, the rigorous biochemistry needed to validate the function of a crosslinked dimer and avoid artifacts associated with chemical crosslinking extend beyond the scope of this study.

4. The major argument for the effects being dimerization dependent are from the results of the D51R mutation, as it is interpreted that this mutation solely affects dimerization. However, it seems plausible that other things may change in the protein that impact catalytic activity that are outside of the dimerization. It is unclear as to whether this particular mutant is well folded on its own, thus further analysis of the protein would be useful here. Is the binding and catalytic activity preserved for D51R in solution, and how does this compare to wild-type under monomeric conditions? If dimerization was observable, for example via AUC, then this would also provide supportive evidence of the fold, as well as report the change in stability.

Previous biochemical analysis of zPIP5KA using SEC/MALS showed that this lipid kinase can form a dimer in solution, while zPIP5KA (D84R) was an apparent monomer by SEC/MALS (Hu et al. 2015, Nature Comm., doi.org/10.1038/ncomms9205). In our hands, we observe the same SEC elution profile for zPIP5KA (wild type and D84R) that was reported by Hu et al. (2015) (see Figure 1 —figure supplement 1). Hu et al. (2015) also previously showed by circular dichroism that mutations that disrupt the zPIP5KA dimer interface do not change the α helical character of the kinase compared to the wild type enzyme. This suggests that the kinase is folded properly.

Using smTIRF, we show that Ax647-PIP5KB(D51R) at a low membrane surface density still binds cooperatively to PIP_2_ lipids (Figure 3D). Dwell time analysis of both PIP5KB (WT and D51R) yielded indistinguishable results (Figure 3C-3D and Table 2). This indicates that the D51R mutant is folded properly, because misfolding would disrupt membrane docking and association with PIP_2_ lipids.

Another strong indication that the D51R mutation only disrupts dimerization is our ability to introduce a structural compensatory mutation (D51R/R254D), which fully restores kinase activity (Figure 5A-5D). If the D51R mutation caused misfolding, the lipid kinase activity would still be reduced in the D51R/R254D mutant.

Throughout the course of this study, we empirically determined that mutations that disrupt protein folding cause PIP5K to be proteolyzed during expression and purification from insect cells. Prior to the Hu et al. (2015) zPIP5KA crystal structure, Lacalle et al. (2015, FASEB Journal, doi: 10.1096/fj.14-264606) reported that they determined the dimer interface of PIP5K based on a structural homology model built from PIP4K. However, in light of the Hu et al. (2015) structure paper, it’s clear that the Lacalle et al. model for PIP5K dimerization is incorrect. Prior to this knowledge, however, we purified the following mutants from insect cells based on Lacalle et al. (2015) PIP5K dimerization model: F190A/L191A/P192A and L35A/Y39A/N43A. Both of these reported “dimer interface” mutants were unstable and proteolyzed during the purification, presumably due to protein misfolding (data not shown). We were never able to biochemically characterize any of “dimer interface” mutants reported by Lacalle et al. (2015). However, the Hu et al. (2015) crystal structure of zPIP5KA, allowed us to successfully purify the following dimer mutants from insect cells without any proteolysis issues: zPIP5KA (D84R), hPIP5KB (D51R), and mPIP5KA (D92R).

5. The binding curves for PIP5KA to the membrane in Figure 2F are very clear and present interesting results on the different binding capacities as a function of PIP2. Since the lower leg of the monomer-oligomer equilibrium reaction on the membrane (Figure 4D) will be dependent on the membrane associated protein density, it is important to have this full binding relationship for all constructs in order to interpret the results from the different experiments. As such, this information seems essential for the interpretation of the D51R data. While the dwell time and diffusion distributions show similar behavior to WT, this only reports on the behavior of the molecules that are bound and are all likely to be monomers under the examined conditions (Figure 3C,D). It does not directly report on whether there is the same amount of D51R vs. WT on the membrane. In this case, the data presented in Figure 5, is not expected to have the same amount of membrane bound PIP5K, which could account for the differences in behavior rather than dimerization. Indeed, the addition of 1 μm of D51R increases the observation of the compositional patterns.

We thank for the reviewer for this excellent observation. In our revision, we include a plot that shows the relationship between mNG-PIP5KB solution concentration and membrane binding intensity for both wild type and dimer interface mutant (D51R) (Figure 5F). In addition, we reconstitute the bistable spatial patterning reaction in the presence of mNG-labeled PIP5K (wild type and D51R). We find that PIP lipid compositional pattern created by PIP5KB (D51R) contain 2x more kinase bound to the membrane compared to the wild type kinase (Figure 6B-6D). However, the reduced activity of the dimer mutant prevents mutant kinase from robustly competing against the opposing lipid phosphatase to form a stable domain of PI(4,5)P_2_ lipids.

Reviewer #2 (Recommendations for the authors):1. It is not a weakness but rather a limitation that the study is not extended experimentally into real cell situations and only speculates about the potential impact of its finding to the control of the enzymes in real cells.

We agree and look forward to performing complementary experiments in vivo to better understand the mechanisms controlling plasma membrane recruitment of PIP5K in the future.

2. Some of the methods and terms should be defined and explained for the general reader: e.g. It would be useful to define "dwell time" or explain what the Y axis is in Figure 2B. (for example what is CDF?).

In our revision, we defined terminology related to our single molecule experiments. For example, CDF is an abbreviation for cumulative distribution frequency. As suggested by the reviewer, we have defined abbreviations in the main text and figure legends. Additional discussion of related terms is also included in the methods section.

3. Similarly, the detailed description of how the catalytic activity of the enzymes were calculated (lines 270-310) is important for the experts. However, it would be desirable to "translate" the following sentences to the general reader."When the derivative of the PIP5K kinase reaction traces were plotted against the reaction coordinate, x, the curve displayed a high degree of asymmetry that required a second-order term to fit (kWT(x) = k0 + k1x + k2x2) (Figure 4E). In contrast, the dx⁄dt curve for the dimer mutant was parabolic and could be fit using an equation that describes an enzyme with first order positive feedback (kD51R(x) = k0 + k1x) based on product binding (Figure 4E)." This may not be necessary if the paper is published in a biophysical journal, but for most of the readers of eLife, this description is not very accessible.

We thank the reviewer for point out how our writing is less accessible to most *eLife* readers. We have replaced some of the jargon with terms that communicate a more simple and clear explanation.

4. The legend to Figure 4 Supplement 1. does not really describe what is measured here. First of all, what is measured as "auto-correlation" should be explained. It is not clear what the different colors mean (the concentrations refer to what component?). Panel A-C appear to show graphs on the various PIP5KB forms (wt or mutants), but the Figure legend for panel A does not refer to PIP5KB.5. The same is true for the legend to Figure 4 Supplement 2. From the description, one would expect to see three similar measurements for the three forms of the enzyme. Clearly, panel A shows a curve that is not explained. Panels B and C show FCS density as a function of TIRF intensity. What is measured here (what is FCS density?)? How do these curves inform about lipid kinase activity? Clearly, the Authors should do better to explain what is shown here and how does that translate to kinase activity.

We apologize for the error, which caused confusion. In our original submission, figure legends for Figure 4 —figure supplement 1 and 2 were swapped. We have corrected this error. Additional text has been added to define and explain the meaning of auto-correlation.

6. I think the DrrA designation of what is better known as SidM should at least be clearly stated somewhere (preferably also in the Figure legends).

In the methods sections, we note that this PI(4)P lipid binding domain, DrrA/SidM (544-647aa), is derived from *Legionella pneomophila*. When this PI(4)P sensor is first introduced in legend Figure 6, we now mention that DrrA is also known as SidM in *L. pneomophila.* To be consistent with our previous publication (Hansen et al. 2019 PNAS), we refer this PI(4)P lipid binding domain as DrrA throughout the manuscript. We have also included an expended description in the methods section titled, “Molecular Biology.” The following statement has been added to the main text of the manuscript when DrrA is first mentioned:

“To test this hypothesis, we compared the ability of PIP5KB and the PIP5KB (D51R) dimer interface mutant to form bistable PIP compositional patterns in the presence of an opposing 5-phosphatase with engineering positive feedback based. This was achieved by fusing a minimal PI(4)P binding motif, referred to as DrrA (or SidM/DrrA), to OCRL (Hammond, Machner, and Balla 2014; Zhu et al. 2010). The resulting chimeric 5-phosphatase, DrrA-OCRL, was previously shown to exhibit positive feedback based on PI(4)P recognition (S. D. Hansen et al. 2019).”7. Remove the prime from all "5'-phosphatase". There are no prime numbers on the inositol ring just simple numbers designating the positions on the single carbon ring.

Thank you for the feedback. We removed all the prime numbers when mentioning 5-phosphatase in the manuscript.

Reviewer #3 (Recommendations for the authors):1. Line 150 and Figure 1B: The authors are referring to a dimer mutant of PIP5K. They need to clarify if that means that the protein is locked in its dimerized state or if the dimerization interface is disrupted and thus the protein resides in its monomeric state. If, as indicated by the 'dimer mutant' reference the protein should be locked in a dimer state, why does the SEC elution profile and their annotation rather indicate that the protein elutes as a monomer? Also, it would be beneficial for the reader to indicate the name of the dimer mutant in the text.

All dimer mutants presented in our study disrupt the dimer interface resulting in a monomeric form of PIP5K. The dimer interface mutations are based on the published crystal structure and biochemical characterization of zebrafish PIP5KA (Hu et al. 2015, Figure 3A). In Figure 3B we show a sequence alignment and highlight the specific aspartic acid and arginine residues that form a salt bridge that stabilizes the dimer interface. Building off the work by Hu et al. (2015, Nature Communications) we characterized the functional significance of PIP5K dimerization on membranes and role it serves in regulating catalysis.

2. Figure 1 Supplement 2: Monomer/dimer distribution of mNG-PIP4K in Figure 1E are considerably different to the ones shown for the Alexa488-PIP4K version in supplement 2. Is there a reason for this comparison, especially as the authors are later introducing another chemical label with the Alexa647-conjugation construct? They should rather repeat the experiment with the Alexa647-conjugation construct that they are using later on.

The difference in molecular brightness distributions between Figure 1F-G and Figure 1—figure supplement 3 (note updated figure numbers) reflects the difference in labeling strategies and their respective efficiencies. The main point of this supplemental figure was to emphasize that PIP4KB and PIP5KB can be labeled with different fluorescent probes and we still observe a difference in their molecular brightness distribution. We include this control to emphasize that the difference in distributions shown in Figure 1F-1G is not solely dependent on using mNeonGreen.

In this study, we did not quantify the molecular brightness distribution of Ax647-PIP4KB or Ax647-PIP5KB. Although labeling with Alexa647 results in a very photostable fluorescent protein (see Figure 2—figure supplement 1 for photobleaching kinetics), the labeling efficiency achieved using Sortase mediated peptide ligation is 25-45% in the presence of Alexa647. We consistently observe this low labeling efficiency independent of the specific protein being labeled. We believe that Sortase generally has a difficult time using the Alexa647-LPETGG peptide as a substrate. Interestingly, Alexa488-LPETGG results in a higher labeled efficiency (50-90%) for all proteins we’ve labeled to date. However, the brightness and photostability of Alexa488 is comparable to mNeonGreen. In this study, we found that mNG kinase fusions were the more reliable option for measuring molecular brightness, because we consistently obtain ~80% mature and fluorescently labeled enzyme. Our revised manuscript notes the measured labeling efficiency in the figure legend. We also include the following discussion the main text of the manuscript concerning the important point raised by the reviewer:

“We also observed similar differences in molecular brightness comparing recombinantly expressed and purified PIP4KB and PIP5KB that were chemically labeled with Alexa488 in vitro using Sortase mediated peptide ligation (Figure 1—figure supplement 3). However, the lower labeling efficiency achieved using Sortase mediated peptide ligation, led us to measure the molecular brightness distribution of PIP5K homologs and paralogs using exclusively mNG fusion proteins in this study.”

3. Figure 1B: In Figure 1B, the 'dimer' mutant of PIP5K is annotated as zPIP5KA(D84R), however in the respective figure caption, it is referred to as zPIP5KA(D51R) mutant – which one now is the correct mutant? The authors should use a size standard to calibrate the SEC elution profiles to ensure the correct detection of monomers and dimers (SEC-MALS would be even better suited). The critical reader would furthermore like to see an SDS-PAGE gel of the purified proteins, as especially the elution profile of PIP5K indicates impurities.

We corrected this typo to indicate the correct dimer interface mutation of D84R for zPIP5KA. For reference, Figure 3B shows a sequence alignment of PIP5K protein family members with the amino acid side chains that stabilize the dimer interface.

In our revised manuscript, we include data showing the calibration of our Superdex 200 column using a variety of protein molecular weight standards (Figure 1—figure supplement 1). In legend Figure 1—figure supplement 1, we note that the elution profile represents the final step in the zPIP5KA protein purification. We also indicate the presence of small molecular weight impurities in the SEC elution profiles using asterisks (Figure 1—figure supplement 1C).

In our revised manuscript, we also include new a SEC elution profile for purified mNG-PIP5KB and mNG-PIP5KB (D51R) in Figure 4A. The elution volume for protein molecular weight standards are indicated in the graph.

Our typically yield of 0.2 mg total protein from 4 liters of insect cells prohibiting extensive characterization of oligomerization using SEC. Given that SEC alone does not report absolute molecular weight of protein complexes, we prioritized the more definitive single molecule TIRF experiments to characterize the oligomerization state and dynamics of membrane bound mNG-PIP4K and mNG-PIP5K.

4. Figure 1C/D: The authors should indicate the abbreviations of both proteins in the respective figure captions.

In the main text, we include the following statement:

“Comparing the molecular brightness distributions of mNG-PIP5KA (mouse), mNG-PIP5KB (human), mNG-PIP5KC (human), mNG-zPIP5KA (zebrafish), and mNG-yMss4 (yeast multicopy suppressor of Stt4 or yeast phosphatidylinositol 4-phosphate 5-kinase), we observed only single peaks corresponding to monomeric kinases labeled with a single fluorescent mNG (Figure 1I).”

In Figure legend 1, we define abbreviation shown in Figure 1 as follows:

“Molecular brightness frequency distribution plots measured in the presence of mNG tagged mouse PIP5KA (5KA), human PIP5KB (5KB), human PIP5KC (5KC), zebrafish PIP5KA (z5KA), yeast Mss4 (yMss4), and human PIP4KB (4KB).”

5. How do the authors account for bleaching in their TIRF single particle tracking assay?

In our revised manuscript, we report the bleaching kinetics of surface immobilized Alexa647 labeled PIP5K (t _bleaching, Ax647-5K_ = 26.7 seconds). This data is reported in the main text, Figure 2 legend, and plotted in Figure 2 —figure supplement 1. Rather than correcting all our dwell time measurement, we note that the bleaching kinetics are significantly long compared to the single molecule membrane interactions.

6. Supplementary Figure 1A: What is the reproducibility of the mNG-calibration curve for the quantitative analysis of kinase presence in the cell lysate? As indicated in the figure caption, only a representative curve is shown, however adding standard errors for each concentration and the indication of the number of replicates would need to be included.

In our revised manuscript, three mNG-calibration curves are now shown in Figure 1 —figure supplement 2. Although our fluorescent protein plate reader measurements are pretty consistent day-to-day, for these measurements we generate a new mNG-calibration curve every time we measure the concentration of mNG-PIP5K in cell lysate. This is similar to creating a BSA standard curve every time a Bradford assay is performed. As a result, adding standard error bars for each concentration was unnecessary.

7. Lines 180-181: Several mNG-fusion constructs are named, but it is completely unclear for the reader what they are, as they have not been introduced (e.g. mNG-PIP5KA/B/C or mNG-yMss4?). Even more importantly, in Figure 1E, which is the figure that has been referenced in the text, there are again other abbreviations for the used constructs that do not fit the ones named in the text or have not been referenced in the text. Likewise, also no consistency between the abbreviations between Figures 1E and F. the authors need to clarify this and keep the names of constructs consistent throughout the whole text, figures, and supplementary information.

We thank the reviewer for recognizing the inconsistencies in our naming of the PIP5K paralogs and homologs. We have corrected the inconsistencies found in Figure 1 and throughout the main text, figures, and supplementary information.

In the main text, we include the following statement:

“Comparing the molecular brightness distributions of mNG-PIP5KA (mouse), mNG-PIP5KB (human), mNG-PIP5KC (human), mNG-zPIP5KA (zebrafish), and mNG-yMss4 (yeast multicopy suppressor of Stt4 or yeast phosphatidylinositol 4-phosphate 5-kinase), we observed only single peaks corresponding to monomeric kinases labeled with a single fluorescent mNG (Figure 1I).”

In Figure legend 1, we define abbreviation shown in Figure 1 as follows:

“Molecular brightness frequency distribution plots measured in the presence of mNG tagged mouse PIP5KA (5KA), human PIP5KB (5KB), human PIP5KC (5KC), zebrafish PIP5KA (z5KA), yeast Mss4 (yMss4), and human PIP4KB (4KB).”

8. Figure 1F: The authors are referring to a step size distribution in the figure caption, but at the same time state diffusion coefficients for the respective constructs – they need to elaborate on how exactly they got the diffusion coefficients from the step size distribution, more specifically: what kind of fitting they used. As in Figure 2C, the fit should also be included in the plot. Furthermore, what is the error and standard deviation for the diffusion coefficients? It would be beneficial for the reader to also state the deviations in lines 190/191.

In our original submission, the method used to fit the step size distribution was described in the methods section titled, “Single particle tracking.” Below are the equations used to fit the distributions throughout the manuscript:

Single species model:

f(r)= r2Dτe−(r24Dτ) Two species model: f(r)= αr2D1τe−(r24D1τ)+(1−α)r2D2τe−(r24D2τ) In the case of Figure 2C, the dashed black line represents the curve fit of the step size distribution. In our revised manuscript, we include the appropriate statistics and errors associated with our diffusion coefficient measurements. In Figure legend 2C, we state that the dashed black line represents the curve fit used to calculate the diffusion coefficient.

9. As stated in line 192, the authors are comparing the diffusion coefficients of the cell lysate kinases with purified kinases in Figure 2C. However, in the two cases different membrane compositions have been used – in Figure 1F they state that 2 % PIP2 have been used, while in Figure 2C 4 % have been incorporated. Why do they change the proportion of PIP2 in the membrane between the two cases? This inconsistency especially impedes their assumption in line 193, that the membrane binding of kinases in the cell lysate is indistinguishable from the purified case.

Thank you for pointing out that we changed two variables when making the comparison between the mNG-PIP5K in lysate and purified proteins. To clarify, the data in Figure 1 was collected using SLBs with 4% PI(4,5)P_2_, while the membranes in Figure 2 contain 2% PI(4,5)P_2_. The rationale for lowering the 2% PI(4,5)P_2_ was mainly to observe the protein density dependent change in Ax647-PIP5KB membrane dwell time (Figure 2B) without encountering problems associated with photobleaching and underestimating the change in the dwell times.

In our revised manuscript, we provide broader characterization of PIP5K membrane diffusivity. Our conclusion, is that diffusivity of PIP5K is not significantly different in the presence of 2% versus 4% PI(4,5)P_2_. In addition, mNG-PIP5K visualized dilute cell lysate also behave similar to purified mNG-PIP5K. Please refer to Table 1, Table 2, and Figure 3 —figure supplement 1 for additional quantification and statistics from single particle tracking data.

10. Figure 2B: For readers that are not familiar with single particle tracking, the cumulative density function should be introduced in the caption of the figure or even in the axis label. Same comment for Figure 3CDE.

We have included a more accessible description of cumulative distribution frequency (CDF).

The methods section includes the following description:

“To calculate the single molecule dwell times for Ax647-PIP4K and Ax647-PIP5K we generated a cumulative distribution frequency (CDF) plot using the 50 ms frame interval as the bin size. The log_10_(1-CDF) was then plotted against the dwell time and fit to either a single or double exponential decay curve.”

We included the following statement in Figure Legend 2B:

“Ax647-PIP5K dwell times were calculated by fitting log_10_(1-cumulative distribution frequency (CDF)) to either a single or double exponential decay curve (dashed black lines). Bin size equals 50 ms. See Table 1 for values and statistics.”

11. Figure 2C: Again, a new abbreviation is being introduced for PIP5KB.

We have corrected the abbreviation in Figure 2C to be more consistent throughout the manuscript.

12. Figure 2F: Error bars are missing and SEM of Kd values. In the respective caption, they should state the number of independent replicates. They also need to indicate in the caption that the nh – values they are referring to in the figure are the Hill coefficients. The manuscript would benefit considerably if they would repeat the experiments shown in this Figure also with the D51R mutant – especially as they are comparing the mutant with the wild type throughout the manuscript, and only for Kd values the respective differences cannot be judged. Similar comment for Figure 4AB regarding error bars and replicate numbers.

We have included the statistics and additional information of the curve fitting in Figure 2F. In the main text, we refer to the curving model as the “concerted model for cooperativity,” which is the terminology used to describe the Hill equation. We added the following text to Figure legend 2:

“Lines represent curve fit using concerted model for cooperativity (i.e. Hill equation). n_H_ is the Hill coefficient. Points are mean values [N = 15-20 fluorescent intensity measurements per sample from 1 technical replicate].”

As suggested by the reviewer, we provide new experimental data comparing the membrane binding behavior of mNG-PIP5KB and mNG-PIP5KB (D51R). At saturating concentrations, we observed a 2-fold difference in protein surface density comparing mNG-PIP5KB and mNG-PIP5KB (D51R). This is consistent with a model proposed by Amos et al. (2019, Structure, doi.org/10.1016/j.str.2019.05.004) that is based on molecular dynamic simulations that showed only a single subunit of a PIP5KA dimer can bind to the membrane at a time.

The kinetic data in Figure 4 (know called Figure 5) was significantly updated to include the proper statistics. The following text can be found in Figure legend 5:

“(A-C) Representative kinetic traces monitoring the production of PI(4,5)P_2_ in the presence of 1-5 nM PIP5KB, PIP5KB (D51R), or PIP5KB (D51R/R254D). The PI(4,5)P_2_ membrane surface density was estimated based on the membrane localization of 20 nM Ax488-PLCd. Initial membrane composition: 96% DOPC, 4% PI(4)P. (D) Quantification of reaction half-time from trajectories shown in (A-C). Bars equal mean values [N = 2-4 reactions per concentration, error = SD].”

13. Figure 3CD: The authors need to indicate in the caption or the figure that this is a low protein density condition.

Figure legend 3 has been revised to state:

“(C-D) Dimerization is not required for cooperative PI(4,5)P_2_ binding. Representative single molecule dwell time distributions measured at low protein density in the presence of either (C) 1-5 pM Ax647-PIP5KB and (D) 1-5 pM Ax647-PIP5KB (D51R).”

14. Figure 3F: Please state the respective diffusion coefficients you have obtained from the used fit. Same comment for Supplementary Figure 3 (and adjust x-axis label to be consistent with similar graphs).

The diffusion coefficients, along with statistics (i.e. number of trajectories, number of steps, and error) are included in both the figure legends and newly added Table 1 and Table 2.

15. Supplementary Figure 4 (Figure supplement 1 and 2): It seems like the Figure captions of Supplementary Figure 4 Supplement 1 and 2 have been mistaken for each other – please readjust.

We apologize for the error. We accidently swapped figure legends for Figure 4—figure supplement 1 and 2 in our original manuscript.

16. Supplementary Figure 4 (Figure supplement 1): In panel B apparently different protein concentrations have been used compared to panels A and C – why?

We originally performed lipid phosphorylation reactions at different concentrations of PIP5K (WT and D51R) because the proteins have different levels of activity. In our revised manuscript, we include kinetic traces for the lipid phosphorylation reactions, as well as the reaction half-times (Figure 5A-5D). All experiments have been performed at similar concentrations for wild type and mutant PIP5KB to more easily compare the reaction half-times.

17. Supplementary Figure 4 (Figure supplement 2): Panels BC – Please indicate error bars and replicate numbers.

We repeated these experiments comparing similar protein concentrations. The standard deviation for our reaction half-time measurements are now plotted in Figure 5D.

18. Figure 4A: Why don't the authors compare the reaction half-times of wild-type and the D51R mutant at similar concentrations to enable a more direct comparison between the two? Are the reaction half-times based on the kinetic traces seen in the respective supplementary Figure? If yes, why is there such a difference in the reaction half-time between the trace at 1 nM PIP5K in the supplement (roughly 160 seconds) and the 280 seconds stated in the main Figure? Why are the traces for higher PIP5K concentrations not included in the supplement?

Thank you for identifying this poorly presented data. In our revision, we include data that directly compares reaction kinetics of lipid phosphorylation in presence of PIP5K, wild type and mutants, measured at the same protein concentrations (Figure 5A-5C). Quantification of the reaction half-times for multiple kinetic traces is now included in Figure 5D.

19. Figure 2 and Figure 4: Why is the fluorescence of a membrane dye used to calibrate for PIP5K membrane density in Figure 2, whereas a more sensitive FCS-approach is used in Figure 4 to do the same? They should have likewise measured FCS for the Alexa-fusion construct in Figure 2.

In our manuscript, we present two complementary approaches to quantify the surface density of PIP5K. The use of a fluorescently membrane dye is an established method for quantifying membrane surface densities (see Galush, Nye, and Groves 2008). It has similar sensitivity to FCS measurements.

The slow diffusivity of Ax647-PIP5KB and mNG-PIP5KB prevented us from directly performing FCS measurements on membrane bound fluorescently labeled PIP5KB. Instead, the FCS data included in our manuscript is a measure of mNG attached to a supported lipid bilayer. The fluorescence intensity and membrane density of mNG was then used to calibrate the fluorescence intensity of other mNG fusion proteins (e.g. mNG-PIP5KB).

Due to limitation in instrumentation, we were unable to perform additional FSC measurements during the period of the manuscript revision. However, we don’t believe this is critical for interpreting our data and it does not change the conclusions of our manuscript.

20. Lines 294-301: in Line 296, the authors give a definition of the reaction coordinate x. However, they are not defining the omega factor that they are using, and it is not clear what they mean with PIP1 and PIP2 – should that be the educt and product? Please clarify and annotate in the text.

The omega factor refers to the fraction of either PIP1 or PIP2. For clarity, PIP1 and PIP2 have been redefined in the text to be PI(4)P and PI(4,5)P2, respectively. The following text was added to the main text:

“To analyze the feedback profiles, we plotted the rate of PI(4,5)P_2_ production, dxdt, as a function of the reaction coordinate, x, as x ≡σPI(4,5)P2/(σPI(4,5)P2+σPI(4)P). σ denotes the membrane density of each PIP lipid species throughout the entire reaction trajectory.”

21. Lines 324-325: Please add the information that the DrrA-OCRL construct is the opposing 5`-phosphatase.

In the methods sections, we note that this PI(4)P lipid binding domain, DrrA/SidM (544-647aa), is derived from *Legionella pneomophila*. When this PI(4)P sensor is first introduced in legend Figure 6, we now mention that DrrA is also known as SidM in *L. pneomophila.* To be consistent with our previous publication (Hansen et al. 2019 PNAS), we refer this PI(4)P lipid binding domain as DrrA throughout the manuscript. We have also included an expended description in the methods section titled, “Molecular Biology.” The following statement has been added to the main text of the manuscript when DrrA is first mentioned:

“To test this hypothesis, we compared the ability of PIP5KB and the PIP5KB (D51R) dimer interface mutant to form bistable PIP compositional patterns in the presence of an opposing 5-phosphatase with engineering positive feedback based. This was achieved by fusing a minimal PI(4)P binding motif, referred to as DrrA (or SidM/DrrA), to OCRL (Hammond, Machner, and Balla 2014; Zhu et al. 2010). The resulting chimeric 5-phosphatase, DrrA-OCRL, was previously shown to exhibit positive feedback based on PI(4)P recognition (S. D. Hansen et al. 2019).”

22. Figure 5B: What is meant with 0`-9`? Are these frame numbers (if yes what is the time difference between them? If the numbering should indicate time – what unit is it? Same question for Figure 6A).

We apologize for the confusing units. In our revised manuscript, we clarified the units to indicate time in minutes.

[Editors' note: further revisions were suggested prior to acceptance, as described below.]

The manuscript has been improved but there is one remaining issue that needs to be addressed, as outlined below:Regarding the text starting on line 375:"Conversely, the phosphorylation rate constant for wild-type PIP5KB began with a slow rate and then transitioned to a rate of 2.7x10-3 lipids/μm2•sec per kinase as the reaction progressed."However, when examining the plots in Figure 5G, it is not clear that there is a slow component to the rate. If the slower rate is in the region of the graph close to the origin, then this area should be highlighted and zoomed in with an inset plot showing an expanded scale. Otherwise, this statement is not corroborated with the data as shown.

Thank you for the feedback concerning Figure 5G. This is a challenging piece of data to present. Based on your comment, we have modified Figure 5G to more clearly explain the transition in reaction kinetics shown in this plot. In addition, we have modified the main text and Figure Legend 5 to more clearly explain these results. An updated version of the manuscript and Figure 5 has been uploaded to reflect the new changes.

Main text edits (lines 375-381):

“Conversely, the phosphorylation rate constant for wild-type PIP5KB began with a slow rate and then transitioned to a rate of 2.7x10-3 lipids/μm2•sec per kinase as the reaction progressed (Figure 5G, see arrows indicating the transition from slow to fast kinetics for wild type PIP5KB). After the reaction reached the maximum velocity, the kinetics gradually slowed down due to substrate depletion. Overall, the difference in per-molecule reaction kinetics comparing PIP5KB and PIP5KB (D51R) was consistent with dimerization enhancing lipid kinase activity (Figure 5E), and establishes a positive feedback mechanism.”

Figure legend 5G edits (lines 755-762):

“…(G) Membrane-mediated dimerization enhances PIP5KB catalytic efficiency independent of membrane localization. Membrane localization of mNG-PIP5KB (WT and D51R) and production of PI(4,5)P2 were simultaneously monitored on supported membranes. Each data point in the plot represents the instantaneous velocity expressed as the number of PI(4,5)P2 lipids generated per μm2 per second per kinase as a function of the substrate density. The reaction catalyzed by PIP5KB displays the following three phases: (1) slow kinetics at reaction start, (2) rise to maximum reaction velocity, and (3) gradual decline in reaction velocity due to substrate depletion. (H)…”